# Neo-epidemiological machine learning based method for COVID-19 related estimations

**Mouhamad Bodaghie**[1]*, **Farnaz Mahan**[1], **Leyla Sahebi**[2], **Hossein Dalili**[3]

**1** Computer Science Department, University of Tabriz, Tabriz, Iran, **2** Maternal, Fetal and Neonatal Research Center, Family Health Research Institute, Tehran University of Medical Sciences, Tehran, Iran, **3** Breast Feeding Research Center, Family Health Research Institute, Tehran University of Medical Sciences, Tehran, Iran

* m.bodaghi@tabrizu.ac.ir

**Data Availability Statement:** All relevant data can be found here: https://osf.io/ec3df/?view_only=993ade4bf08d4c65988ff0d8d69fb1ad.

**Funding:** There is no funding supporting the present study.

## Abstract

The 2019 newfound Coronavirus (COVID-19) still remains as a threatening disease of which new cases are being reported daily from all over the world. The present study aimed at estimating the related rates of morbidity, growth, and mortality for COVID-19 over a three-month period starting from Feb, 19, 2020 to May 18, 2020 in Iran. In addition, it revealed the effect of the mean age, changes in weather temperature and country's executive policies including social distancing, restrictions on travel, closing public places, shops and educational centers. We have developed a combined neural network to estimate basic reproduction number, growth, and mortality rates of COVID-19. Required data was obtained from daily reports of World Health Organization (WHO), Iran Meteorological Organization (IRIMO) and the Statistics Center of Iran. The technique used in the study encompassed the use of Artificial Neural Network (ANN) combined with Swarm Optimization (PSO) and Bus Transportation Algorithms (BTA). The results of the present study showed that the related mortality rate of COVID-19 is in the range of [0.1], and the point 0.275 as the mortality rate provided the best results in terms of the total training and test squared errors of the network. Furthermore, the value of basic reproduction number for ANN-BTA and ANN-PSO was 1.045 and 1.065, respectively. In the present study, regarding the closest number to the regression line (0.275), the number of patients was equal to 2566200 cases (with and without clinical symptoms) and the growth rate based on arithmetic means was estimated to be 1.0411 and 1.06911, respectively. Reviewing the growth and mortality rates over the course of 90 days, after 45 days of first case detection, the highest increase in mortality rate was reported 158 cases. Also, the highest growth rate was related to the eighth and the eighteenth days after the first case report (2.33). In the present study, the weather variant in relationship to the basic reproduction number and mortality rate was estimated ineffective. In addition, the role of quarantine policies implemented by the Iranian government was estimated to be insignificant concerning the mortality rate. However, the age range was an influential factor in mortality rate. Finally, the method proposed in the present study cofirmed the role of the mean age of the country in the mortality rate related to COVID-19 patients at the time of research conduction. The results indicated that if sever quarantine restrictions are not applied and Iranian government does not impose effective interventions, about 60%

**Competing interests:** The authors have declared that no competing interests exist.

to 70% of the population (it means around 49 to 58 million people) would be afflicted by COVID-19 during June to September 2021.

## 1 Introduction

Currently, the 2019 newfound Coronavirus (COVID-19) pandemic is the most prevalent global health problem affecting 212 countries and regions worldwide. COVID-19 was reported in China (Wuhan city) for the first time on December 31, 2019 [1]. Approximately 40% to 45% of people with SARS-CoV-2 are asymptomatic and spread in silence among human populations [2]. The fatality ratio highly depends on age distribution. As such in the age group over 85 years this ratio is 14.8% and in less than 10 years is close to zero. On average, the incubation period of COVID-19 infection is four days (however in different situations ranging from 2 to 14 days); 80% of cases are mild or asymptomatic, 15% are severe and require oxygen supply, and 5% are ventilated at hospitals [3]. The World Health Organization (WHO) has developed strategies to reduce the transmission of the virus including intervention to stop person-to-person transmission (the main transmission of virus is person-to-person), identifying asymptomatic individuals and isolating them, establishing disease control laws in the community including quarantine or social distancing, detecting appropriate medications or vaccines, and minimizing social and economic effects on communities [4]. If at a specific point in time ($t$), we define the percentage of "patients" in the population with the symbol $P_t$, the rate of disease transmission is the symbol $C_t$, and the infectious strength of the disease is shown as a function of $f(I)$. The formula 1 shows these three factors [5]:

$$f(I, t) = C_t \cdot P_t \tag{1}$$

The prevalence of the disease and its spread in society, based on the function (1), is always moving towards a balanced pattern. From the point of view of epidemiology, two types of balanced patterns can be defined for the diseases:

1. Disease-free balance in which society is—certainly or almost—disease-free and stable in this respect. e.g. at present conditions, for smallpox a disease-free balance can be defined in human society.

2. pandemic balance where the disease is pandemic (or local) in the community and the occurrence of new cases follows a fixed and expected pattern [5].

In an epidemic, this balance is completely upset. Before the advent of Covid-19, the pattern of disease-free balance could be defined for a given society. But with the onset of the first cases, the pattern lost its balance and a pandemic occurred. This scenario will eventually lead to an pandemic equilibrium pattern. But it is very important when and at what cost this balance will be achieved (from the perspective of basic product number value and mortality rate) [6], also, the type and timing of control measures and interventions will be very important and effective at the end of this story. Estimation of basic reproduction number ($R0$) is important and valuable in monitoring the status of an pandemic until an epidemiological equilibrium is achieved [7]. One of the most important and practical indicators used to show the pattern of expansion of Covid-19 is the basic reproduction number or ($R0$) nought R in which heterogeneities have been seen in different societies with the same conditions and parameters [8]. By definition, it is attributed to sensitive people who have been afflicted by contacting the sick people. This index basically indicates how contagious an infectious a given disease could be [7]. Concerning

epidemiological concepts, the primary case is the person who transmits the disease to a small number of people in the community. This case has transferred the disease to all people during its infectious period. These people are called secondary cases. Each secondary case transmits the disease to other susceptible individuals. This cycle repeats itself periodically and the disease spreads in the community. Direct calculation of the basic reproduction number is possible when in a society with a very sensitive population, the number of people infected with the disease is carefully tracked from the initial case. In this case, the basic reproduction number can be calculated directly by calculating the average number of people that each specific patient can infect [9]. The severity of the disease transfer is not constant at all, and is continuously changing. Therefore, its estimation at the beginning, middle and end of the pandemic can be completely different, but the difference in the same conditions in different societies may be due to a weakness in the model for extracting this index. Given that the basic reproduction number by definition should be calculated based on the course of the disease in a very sensitive community, so the symbol R0 can be used only in calculations or estimations at the beginning of the pandemic. Over time pass the reduced number of susceptible people in a certain society is denoted by the symbol Rt or, for simplicity, by symbol R. Theoretically, basic reproduction number depends on three factors [10–12]:

1. The risk of transmitting the disease per contact depends on two factors: the type of disease and the type of contact. It was first stated that Covid-19 disease is transmitted through direct contact with infected people and surfaces. Observing physical distance in socializing with others can be reduced. Estimations of this risk in routine contacts are estimated at 10 to 20%. But later the danger of infected surfaces was estimated so low that put the estimation results under the question [13].

2. The average number of contacts per person per time unit, which depends entirely on population density, the frequency and manner of travel in the community, people's culture and social indicators can be modified with measures such as restricting or prohibiting meetings, reducing travel numbers, reducing the amount of interactions, isolation of patients or people suspected of having the disease, and quarantine of seemingly healthy individuals.

3. The average period of infectivity, which, although is considered a stable biological indicator, can be reduced by therapeutic interventions and the administration of effective antibiotics. Estimations for Covid-19 disease show that the course varies from 14 days in mild cases (on average 5 days before symptoms and 9 days after) to several tens of days in severe and long cases [14].

In addition to direct calculation, which is not always easily possible, it is possible to estimate the basic reproduction number based on mathematical models or machine learning methods such as neural network by using scientific assumptions and analyzing the time trend of occurrence.

Many pandemics, including the recent Covid-19 pandemic, are possible due to the high percentage of asymptomatic subclinical patients and the expected low numbers in the care system, especially in cases where disease care is generally passive. Identifying primary and secondary cases and calculating the basic reproduction number directly is not easily possible. However, using learning methods and models such as neural networks, it is possible to approximate the basic reproduction number with a reasonable amount of basic error [15].

On the other hand, neural networks combined with approaches having meta-heuristic algorithms in their learning algorithm can work more efficiently in approximating the basic reproduction number, which the present study seeks to prove experimentally. Previous approaches to basic reproduction number used a simple method to approximate it, based on the average

age range of infection with an infectious agent, which is more applicable to diseases that the number is expected to be relatively high for [12]:

$$R_0 = 1 + L/A \qquad (2)$$

In the formula (2), L is the average life expectancy of the population and A is the average age at the time of infection. Obviously, this formula does not have the necessary efficiency and proper mechanism to calculate quarantine conditions that have different ratings.

Another previous approach to estimating the basic reproduction number is to use the rate of disease growth in the population taking into account the latency and infectivity rate of the disease. The basic reproduction number can be estimated by this method using three factors:

1. Average incubation period of the disease

2. Average period of pathogenesis

3. Disease growth in the community. The following formula can be used for calculation [15]

$$R_0 = K^2(LD) + K(L + D) + 1 \qquad (3)$$

In the formula (3), $L$ is the mean latency period, $D$ is the average pathogenicity period, and $K$ is the disease growth rate at the logarithmic scale. To calculate $K$, we can use Formula (4), where $Y$ represents the number of patients and $t$ represents time:

$$K = [\text{Ln}(Y_t/Y_0)]/t \qquad (4)$$

This formula also has the problem of relation 2 which does not provide a mechanism for calculating social interventions and its effect on basic reproduction number.

Today, artificial intelligence approaches, especially machine learning techniques for modeling epidemiological data, have become increasingly common in the literature. These methods have the potential to improve our understanding of health and the opportunities for therapeutic and preventive interventions far beyond past techniques.

The reason for the superiority of artificial intelligence algorithms is that not only they have a high computational speed in the face of large and heterogeneous data, but also they can approximate high-order nonlinear data relationships with high computational complexity, thereby providing new knowledge; such as in epidemiological forecasting of diseases. Hence, they can achieve what the previous methods could not, because the previous methods do not have the computational ability with optimal time for this volume of data and in some cases, do not have suitable computational tools for some specific data (such as descriptive and fuzzy data). For example, when it comes to calculating the impact of government interventions through quarantine control of COVID-19 disease, finding data relationships and discovering knowledge from data that is not clearly articulated (such as the type of quarantine, social groups subject to quarantine, etc.) the previous methods of obtaining the prevalence rate or R number require a clear definition of such data.

The present study proposed a combined neural network algorithm with BTA algorithm to create a supervised machine learning model to estimate the basic reproduction number with taking into account the quarantine conditions at different levels. The error of estimating the basic reproduction number of the proposed algorithm in the present study is compared with combined neural network with the PSO algorithm as another learning algorithm.

Therefore, in general concerning the inefficiency of practices such as 2, 3 for estimating the production rate which can be used to predict the number of disease cases, the followings are suggested:

1. These formula are only useful for extracting the basic reproduction number and cannot be generalized to other indices like morbidity rate, prediction of morbidity and/or prediction of the time of herd immunity.

2. They work comprehensively on deterministic data. As such they lack possible and efficient flexibility concerning non-deterministic data (probabilistic or fuzzy) and the combination of deterministic and non-deterministic data.

3. For the existing noise in data, they do not produce reliable responses.

4. They only produce reliable responses when there is a linear relation between data (variables) and given responses. In addition, they are not applicable for non-linear relations.

5. Addition of any data after a specific response is produced requires remodeling which in turn contributes to increase in computational complexity.

6. Computational complexity of these practices puts their efficiency for macro datasets under the question. For example, the computational complexity of formula 3 at best mode is of $0$ $(n^2)$ degree.

The whole control of the disease is associated with many complications including the lack of a specific medication or vaccine, the relative longevity of the incubation period, the possibility of transmission of infection during the preclinical as well as the subclinical period, and the inability to predict the virus behavior. However, several factors including confirmation of person-to-person transmission, evidence of the possibility of airborne droplet transmission, and high population density, indicate the importance of evaluating transmission mode and the need to predict the type of virus behavior in infection phase. One of the basic ways to control the pandemic is to better understand the virus behavior and its characteristics that might assist making appropriate decisions and effective planning in any given community. Now there is no pattern to predict the trend of the pandemic of COVID-19 in Iran and its epidemiological cycle is very ambiguous. Because concerning the COVID19 pandemic, a large number of patients are subclinical (only carriers of the virus) or have very mild and non-specific symptoms, it is practically impossible to perform a diagnostic test at the general level, therefore, it is not possible to estimate the true number of the patients by conventional statistical methods. As a result, there is no estimation of the growth rate of the disease. On the other hand, due to the emergence of this pandemic and the unknown dimensions of the virus, there is no certainty about the behavior of the virus in different communities, health levels, and weather situations. In the present study, we intended to develop a genetic neural network to estimate morbidity, growth rate, and mortality rate of COVID-19 over a three-month period starting from February 19, 2020 to May 18, 2020 in Iran considering the changes in weather variant and the country's executive policies including social distancing, restrictions on travel, closing public places, shops and educational centers. Compared with machine learning based method, due to limitations in defining their parameters as well as limitations in data type coverage, the conventional methods are not efficient in calculating epidemiological indices such as productivity rate index, mortality rate, predicting the number of cases and deaths, and even herd immunity time. That is, whenever it is necessary to extract vital knowledge from seemingly insignificant data, deterministic methods such as formulas 2, and 3 cannot be held accountable at all. Because these methods only produce reliable results when dealing with deterministic

data of which their accuracy is proven and enough for calculation. But the discovery of knowledge from noisy data and the limited number, is of capabilities of machine learning methods, an example of which, along with its efficiency and optimization, will be introduced in the present study to create an epidemiological model. According to the above said issues and studying similar content, the general gaps existing in the body of literature can be listed as below:

1. All models estimating the basic production number need reliable and precise information.

2. The previous models estimating the basic production number have worked on narrow data, and do not cover large data and those with noise.

3. Parameters of the previous models estimating the basic production number are not flexible and do not work with the same data.

4. Discovery process of epidemiological knowledge in the previous models estimating the basic production number is not possible without the mastery and knowledge of an expert epidemiologist.

The contributions of the present study are as follow:

1. Proposing a model based on artificial intelligence to estimate the basic reproduction number, predict the number of cases, mortality rate, the number of deaths and the time of herd immunity during the pandemic of COVID-19 virus.

2. Combining artificial neural network with human based intelligence algorithm of BTA (Bus Transportation Algorithm) for utilization in the mentioned artificial intelligence model.

3. Comparison of artificial neural network hybrid model combined with BTA algorithm, with neural network hybrid model combined with PSO algorithm in estimating the basic reproduction number, predicting number of cases, mortality rate, and herd immunity time during the pandemic of COVID-19, and providing the experimental proof of the model proposed in the present study.

In section 2, in addition to an overview of recent studies aimed at proposing epidemiological models based on artificial intelligence, the existing gaps in related literature were addressed. Section 3 elaborates on the materials and methods; in section 3.1 a new epidemiological model based on artificial intelligence is introduced; in section 3.2 after a brief introduction of Bus Transportation Algorithm, the practice of combining an artificial neural network with BTA is explained; section 3.3 refers to research data and the way it was accessed; section 3.4 describes the initial experimental quantification in the introduced hybrid neural network. In Section 4, the results of the study rekated to the proposed model will be explained in details after implementation in the MATLAB environment and compared with a similar method. This section includes subsections for estimating the basic reproduction number, predicting the number of cases, mortality rates, the number of deaths, and the time of herd immunity during the pandemic of COVID-19. Section 4 is devoted to discussion of the results and finally section 6 presents the overall conclusion of the present study.

## 2 Related works

Since the beginning of the Covid-19 pandemic, many researchers have tried to use machine learning and artificial intelligence methods to introduce epidemiological models of the disease in order to predict patients' conditions worldwide. During this period, researchers introduced algorithms for epidemiological models that were also newly introduced in machine learning [16–18]. Also, providing epidemiological models can be very important in adopting countries'

health policies. Recently, the process of vaccination has begun in different countries, and therefore the epidemiological forecast of Covid-19 can convince health officials to expedite the immunization of their communities. In a research [19], machine learning methods have been used to present the epidemiological model in Mexico. In this research, neural network, decision tree, Bayesian and SVM methods have been used for epidemiological forecasting. This study provides an epidemiological forecast with a sensitivity of 93.34% and a specificity of 94.30% with data from Covid-19 patients in Mexico. Researchers in [20] presented an epidemiological model based on deep learning. This model has been proposed with high accuracy, lower calculation cost, and less need to use observation data. This study finally showed that it has achieved reliable results for learning its deep learning network with limited samples. Researchers in [21] presented a three-step machine learning strategy for classifying risk across a given country based on countries reporting COVID-19 data. They classified four risk groups based on risk of transmission (cases of COVID-19 per million population), risk of death (deaths per million population from COVID-19) and risk of inability to use a COVID-19 test (COVID-19 test per million population). Corona per million population) for countries worldwide. The four risk groups were labeled: "low", "medium-low", "medium-high" and "high". They used the Stack of Gradient Boosting, Decision Tree, and Stack of Support Vector Machine methods in their classification. The method used by these researchers proved the efficiency of machine learning methods for classifying the mentioned epidemiological indices. In [22], a machine learning method inspired by the least squares model (SIMPLS) has been developed to predict mortality at hospitals. Prediction accuracy is randomly assigned to training sets and validation. SIMPLS-based model is developed by isolating rescued people from survivors. The method was able to predict hospital mortality in patients with COVID-19 with moderate predictive strength ($Q^2 = 0.24$) using training and validation kits, which in turn revealed high accuracy (AUC > 0.85). As in [23] it was declared that the basic reproduction number has been the spotlight of reported information during the last six months. The value of the reproduction number has been used by different range of practitioners in medical science, hospitals, political decision makers and public media to justify the strategies used to take the COVID-19 pandemic under control. The study elaborated on the effectiveness of political interventions using the basic reproduction number of COVID-19 across Europe. The researchers proposed a SEIR epidemiological model with a time varying reproduction number, identified through the use of machine learning method. As they report the basic reproduction number was 4.2±1.69, with maximum value of 6.33 in Germany and Netherlands during the early outbreak of COVID-19. The value decreased to 0.67±0.18, with minimum value of 0.37 and 0.28 by May, 10, 2020 in Hungary and Slovakia. A strong correlation was found between passenger air travel, walking, and transit mobility and the effective basic reproduction number with a time delay of 17.24±2.00 days. The proposed new dynamic SEIR model avails flexibility to estimate outbreak control and the exit strategy to inform the political decision makers and identify global safety solutions. In [23] asserted that long-term forecast of COVID-19 pandemic assists health authorities to determine the transmission features of the virus and take effective steps in prevention and control strategies in advance. They proposed Dynamic-Susceptible-Exposed-Infective-Quarantined (D-SEIQ) model; resulting from due modifications applied to Dynamic-Susceptible-Exposed-Infective-Recovered (SEIR) and integrating machine learning based parameters optimization under epidemiological rational constraints. The model was used in order to predict the long-term cumulative numbers of COVID-19 cases in China from January, 27, 2020. Reports from three different regions in China were selected for model evaluation. The results approved the effectiveness of the model in simulating and predicting the trend of COVID-19 outbreak. The researchers declared that integrated approach of pandemic and machine learning models could accurately forecast the long-term trend of the

COVID-19 outbreak. In addition, the parameters of the proposed model were insightful for analyzing COVID-19 transmission and the effectiveness of related interventions in China. In [24], the researchers investigated the disease spreading behavior along with the basic reproduction number benefiting from susceptible-infected-recovered (SIR) model. They simulated the disease transmission activity by using Monte Carlo simulation and analyzed it benefiting from the design of experiment and neural network. The investigated systems were considered as discrete cells for allocating the agents i.e. system population. Varied sizes and population densities were used to observe the finite size effect, while the infectious period was varied to observe its influence on disease transmission dynamics. Results indicated that the number of agents in each phase as a function of time depended on the whole parameters. The main plot suggested that the basic reproduction number maintained with the increased system size; increased to some extent by increasing the density, and a significance increase was observed with the increase in infectious period. For establishing the relationship among parameters, the researchers resorted to neural network and found out the optimized network architecture to be at 3-28-9-1. They also made use of residual plot analysis to confirm the quality of obtained data. The validity of using multiple modeling/analysis techniques i.e. Monte Carlo, design of experiment and neural network, as the supplementary essential tools for modeling the dynamics of SIR disease spreading scheme, was approved with high level of accuracy for data prediction. In another study [25] in order to predict the pandemic velocity of Covid-19 disease in the United States, an epidemiological model has been proposed using the Bayesian time method and random forest algorithm. The study evaluated three areas of New York, Colorado, and Virginia over 21 days. The results of this study showed that political and social reactions change the speed of the pandemic. The epidemiological prediction of Covid-19 and the manner and speed of its spread in South Korea using deep learning has been the subject of another research [26]. This study compared the results of its research with other epidemiological models and proved the effectiveness of the combined method based on the introduced deep learning. Measurement and prevention of COVID-19 using SIR model and machine learning in intelligent health care in Saudi cities was another research subject [27]. In this study, it was pointed out that knowing the number of sensitive, infected and improved cases every day is significant for mathematical modeling to identify the behavioral effects of the epidemic. The model of the study predicted sensitive, infected and improved cases for the next 700 days. The proposed research system predicted whether COVID-19 would spread or disappears in the long run. The simulation data of this study showed that in Saudi Arabia, the virus may be entirely under control only after June 2021.

Although all of above mentioned studies used machine learning in some way to estimate epidemiological indices, the disadvantage of all of these studies is the lack of a general epidemiological model that utilizes machine learning-based abstraction to estimate epidemiological indices. In fact, all the mentioned models used machine learning in the heart of their classic model, which does not lead to filling all the basic gaps of the classic model. For example, all of these models are non-generalizable to different types of epidemiological data and work for a specific type of data. But the abstraction on which the present study is based introduces a model that is highly generalizable to any type of data and space (both fuzzy and even quantum). This model has other capabilities that will be discussed after the model is introduced.

## 3 Materials and methods

### 3.1 Learning-based epidemiological model

Instead of using a model based on a particular formula with specific variables, which can only represent the linear relationship between epidemiological variables, a learning model can

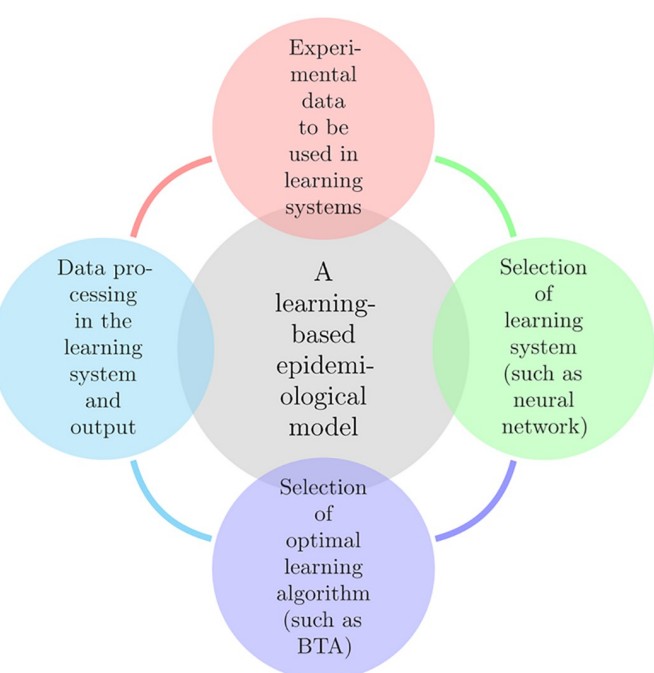

**Fig 1. An epidemiological learning model.**

describe not only the linear and nonlinear relationships of the data, but also can work on deterministic and non-deterministic data (such as fuzzy data) for extracting epidemiological indicators such as: productive rate, predicting the number of cases, mortality rate, predicting the number of deaths and finally the time of herd immunity. This model is a learning cycle that can learn more with each rotation and thus improve its error rate over time. The learning cycle of the epidemiological model is illustrated in Fig 1. This cycle has four stages, each of which is described in detail below.

**3.1.1 Experimental data to be used in learning systems.** Benefiting from previous experiments is an indispensable part of a learning system; where we are fully aware of the input and output data (such as basic reproduction number, prediction of the number of cases, mortality rate, prediction of the number of deaths and the time of herd immunity) and we can consider such experiments as the basis of our system. In the case of COVID-19 pandemic the previous pandemics' data like MERS and EBOLA can be considered as the basis. The results of the (ref 49) could be a proper basis for a learning system when no data is available for estimating the basic reproduction number, prediction of the cases, mortality rate, prediction of the number of deaths and the time of herd immunity. Therefore, if there are any available datasets, it is logical to utilize them.

**3.1.2 Selection of learning system (such as neural network).** A learning system must have appropriate flexibility regarding the data. The learning process must take place with due pace to be able to show required reaction. The architecture of a learning system is usually shaped through test and trial. In the present study the selected learning system is an artificial neural network elaborated on in section 3.

**3.1.3 Selection of optimal learning algorithm (such as BTA).** Any given learning system requires an optimized learning algorithm. The pace of a given learning system highly depends on this algorithm. On the other hand, the value of learning system's error is also dependent on

the algorithm's efficiency. In the present study, BTA was used for learning the neural network which is explained in details in section 3.2.

**3.1.4 Data processing in learning system, and output results.** In each cycle, some outputs are produced that can be used as the input in the next cycle. The process continues to reach an acceptable value of error to make the system more reliable.

## 3.2 Artificial neural network combined with BTA and PSO learning algorithms

Recently, the use of Artificial Neural Network (ANN) for providing epidemiological models has gained much attention [28–30]. Even studies during the COVID-19 pandemic have used neural networks for epidemiological modeling [31, 32]. IIn the present study, we intended to use the artificial neural network combined with Bus Transportation Algorithm (BTA) compared to the artificial neural network combined with Swarm Optimization Learning Algorithm (PSO) [33], to estimate the growth rate of the disease, the total number of patients (including patients with symptoms and without clinical symptoms) with COVID-19. We also considered the role of weather variant and the impact of public operational policies on disease control including the closure of schools, colleges, offices and businesses. It is conceivable to simulate any logical and algebraic function by means of artificial neural networks (ANN) having simple computational elements. Determining the parameters of the neural network unlike biological neural systems, does not pursue any known law so far. The development and functioning of the neural network depends on the learning algorithm of which the network utilizes. Like human biological neural networks with unconventional and inconceivable capabilities that evolves over time and does not initially have these capabilities, a method of learning regulates the neural network over time in such a way that it augments its behavior on the basis of new information and observing current measurements. Therefore, neural networks are complemented by the learning algorithms which can explain the rules of learning. The learning algorithm adjusts the network parameters based on the presentation of the patterns and minimizes the neural networks errors. Based on supervised learning, it is assumed that at each phase of the iteration of the learning algorithm the desired answer of the learning system is prepared in advance. In other words, the learning algorithm has access to the real and desirable answers. That is, the learning algorithm which regulates the parameters of the learning system has access to both the desired and the real answers (obtained from the neural network). In other words, it will have access to learning errors which is the difference between the desired and the actual values [34, 35].

One of the training methods for neural networks is the use of evolutionary algorithms, like PSO and Human- based collective intelligence algorithms such as BTA. The PSO [36], is in the category of optimization methods inspired by the nature of living beings. The recently introduced BTA algorithm falls into the category of Human-based collective intelligence algorithms [37]. All three of these algorithms fall into the larger category of meta-heuristic algorithms [38, 39]. The PSO algorithm is a general method of minimization that can be utilized to deal with problems of which their answer is a point or a level in the n-dimensional space [36]. To benefit from BTA in any optimization problem, at first the bus, stations, and passengers must be certified (definitions inspired by how humans use the bus, and are actually functions that the BTA uses to solve a given problem) and specified to recognize which part of the problem they are. Their related values must also be specified. This is called the BTA modeling. In the next step, we need to have a transportation system that works properly and based on it, we have a continuous optimization. It had been noted that the main idea of the problem solving method by the bus transportation algorithm is implemented according to the amount of loads and unloads of

passengers, and this will continue until the optimal answer is achieved. Stations are auxiliary memories that help us to simplify and organize the problem at every step. As mentioned before, the stations provide an opportunity to evaluate the obtained answers. Passengers are transported so far among the stations in the problem until they reach our desired stations [37]. This shift is based on one or more learning algorithms that monitor how individual and collective implementation works. The neural network combined with BTA is known as ANN-BTA, and the neural network combined with PSO is called ANN-PSO.

## 3.3 Dataset

The data required for this study were obtained through the daily reports of the World Health Organization (WHO), Iran Meteorological Organization (IRIMO) and the census of the Statistics Center of Iran in 2016 which included the following:

- Number of deaths from the outbreak of COVID-19 disease in the period of February 19, 2020 to May 18, 2020in Iran.

- Daily mortality rate based on the number of deaths of patients with COVID-19 in the period of February19, 2020 to May 18, 2020 in Iran.

- The mean age of Iranian population equal to 31.1 provided by the Statistics Center of Iran in the 2016 census.

- The average temperature of Iran's weather in the period of February 19, 2020 to May 18, 2020.

- Temperature difference between the coldest and warmest point of Iran in the period of February 19, 2020 to May18, 2020 in Iran.

- Random values in numerical ranges, [0.1, 0.4], [04, 0, 7], [0.7, 1], [1, 1.3] chosen to estimate disease mortality caused by COVID-19 in the period of February 19, 2020 to May 18, 2020.

The reason for choosing different numerical ranges is that it is possible to provide the best approximation of the mortality rate according to the health and weather variant, and due health policies of Iran's government during the outbreak of the disease caused by COVID-19 and the following time span at the specific period. In the supervised perceptron neural network a, b, c, d, and e were determined as input and f as output. Although data normalization is prevalent when using a neural network, data normalization was avoided because the network outputs were in different ranges. The study data could be accessed through the below mentioned link:

Data of the present study. (https://www.dropbox.com/sh/t224son1u1z5hzq/AAB1GADIi6UdhwUI_C7GLggha?dl=0)

## 3.4 The structure of the used neural network and the evaluation criterion

The network provided the best possible answer with both BTA and PSO algorithms with 3 layers (Input, Hidden, and Output) in the form of 6-5-1. Fig 2 shows the network designed to estimate the mortality rate related to COVID-19 in Iran. In the present study, the aim was to achieve the mortality rate of the disease caused by COVID-19 virus. By achieving the presupposed rate, it would be practically possible to estimate the number of cases affected with the disease between the period of February 19, 2020 to May 18, 2020. Factors such as weather variant and restrictions imposed by the government must also be considered. The first step in designing a multi-layered perceptron neural network is to create network architecture. To create the architecture of the multi-layered perceptron network, the new function of the MATAB

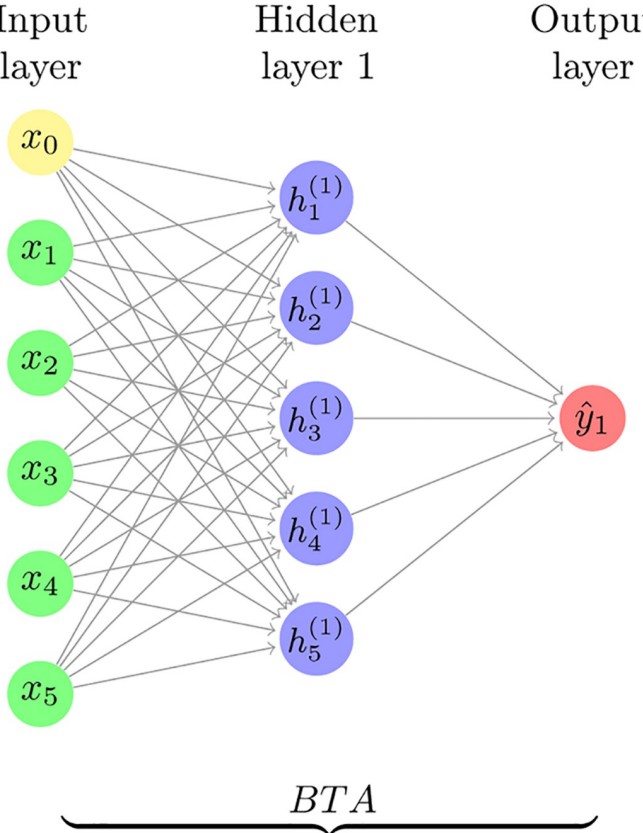

**Fig 2. A neural network designed to estimate the mortality rate of those infected with COVID-19 virus in the period of February 19, 2020 to May 18, 2020.**

2019 software was used. This function has four inputs that connect the created network to the output. The first input is an $R \times 2$ matrix as the minimum and maximum values of each input regarding the $R$ input vectors. The second input is an array of the size of each layer that determines the number of neurons in the different layers. In the present study the amount for 6 input neurons was equal to 5 neurons in the hidden layer and one neuron in the output layer, which is expressed as 6-5-1. The amount of hidden layer neuron was obtained by trial and error, which provided the lowest error rate in both ANN-PSO and ANN-BTA networks. The third input is a cellular array that includes the transfer functions used in each layer, and the fourth input is the name of the used training function which was programmed in the same study.

The fitness function used in the perceptron multilayer neural network is defined as the sum of the mean squared errors in relation 5 [40]:

$$MSE = \frac{1}{N}\sum_{i=1}^{N}(e_i)^2 = \frac{1}{N}\sum_{i=1}^{N}(t_i - a_i)^2 \tag{5}$$

$N$ is the total number of training samples, t is the actual output value and ai is the network output value. As mentioned the BTA and PSO algorithms were used to train the multi-layered

perceptron neural network in MATLAB 2019 software. To train a multi-layered perceptron neural network the data is divided into three sets [40]:

1. Training data

2. Test data

3. Validation data

Training data is the type of data that is used to train the network. The network agents are adjusted according to the errors obtained from this data. Validation data is used to measure the generality of the network. The validation error must be reduced during the training process, such as the training set error. However, when the network attempts to over-match the data the validation error increases. When the validation error for a certain number of repetitions increases, the training is stopped and the amount of weights and biases are matched and coincided to the time at the point that the related error is minimal. The third subset is the test data, which is not applicable during the training process but is used to compare different models. Test data is also used to draw errors in the training data set during the training process. In this study, 80% of the data were used as training data sets and validation data sets; also 20% of the data was used as Test data sets. Regarding the efficiency of the proposed model in the present study the following points can be counted:

1. The method not only is useful for extracting the basic reproduction number, but also it can be generalized to other indices such as mortality rate, prediction of the number of mortality, and/or prediction of herd immunity time.

2. The method not only works on deterministic data, but also has possible efficiency flexibility for non-deterministic data and, combination of deterministic and non-deterministic datasets.

3. Although there is noise in data, reliable responses are produced (the existing noise is detected).

4. If there is linear and nonlinear relation between data and a specific response, the responses are reliably produced.

5. Addition of any data after a given response is produced, does not require remodeling which contributes not to high computational complexity.

6. The computational complexity of the method has required efficiency for macro datasets.

## 4 Results

According to the proposed epidemiological model based on machine learning, which was discussed in detail in the previous section, a hybrid neural network was designed with BTA and PSO learning algorithms. The structure of the designed neural network can be seen in Fig 2 and like the structures of other neural networks, its structure in terms of the number of hidden layers and etc. has been obtained through trial and error. Fig 2 shows the final structure of the ANN-BTA network. Regarding the structure of ANN-PSO network, the same structure has been preserved and it's the only substitute learning algorithm for BTA and PSO; this was presented to compare the proposed method of the present study i.e. ANN-BTA with a similar network, which proves its efficiency in similar problems. After the implementation of ANN-BTA and ANN-PSO networks in MATLAB environment, estimation of the basic reproduction

number and mortality rate due to CIVID-19 pandemic, based on available data for 89 days in the time period of February 19, 2020 to May 18, 2020 in Iran, was investigated. Relying on the two indicators of the basic reproduction number and mortality rate, the number of cases and the number of deaths can be estimated or even predicted. After predicting the number of patients, the time of herd immunity can also be predicted. the results of the present study encompasses three sections. The first section 4.1 refers to testing the network. It is obvious that testing the network cannot be done through classic methods, and needs to be conducted through testing with other netwroks. The second section 4.2. elaborates on the results obtaine concerning the estimation of the basic reproduction number and, Estimating the number of cases, mortality rate, number of deaths and time of herd immunity. The third section 4.3 explains the effecticvness of applied public nterventions.

## 4.1 Testing the ANN-BTA and ANN-PSO networks

In an artificial neural network, the output is reliable when the value of MSE in training and experimental samples is minimum. Although the sum of squared error is less, it confirms that in that range the network was able to perform a proper approximate function. And related ranges can be determined as the mortality rate of the cases infected by COVID-19 according to the input data in the period of February 19, 2020 to May 18, 2020 in Iran. Figs 3–6 show the convergence trend of network in finding the best function between input and output in the

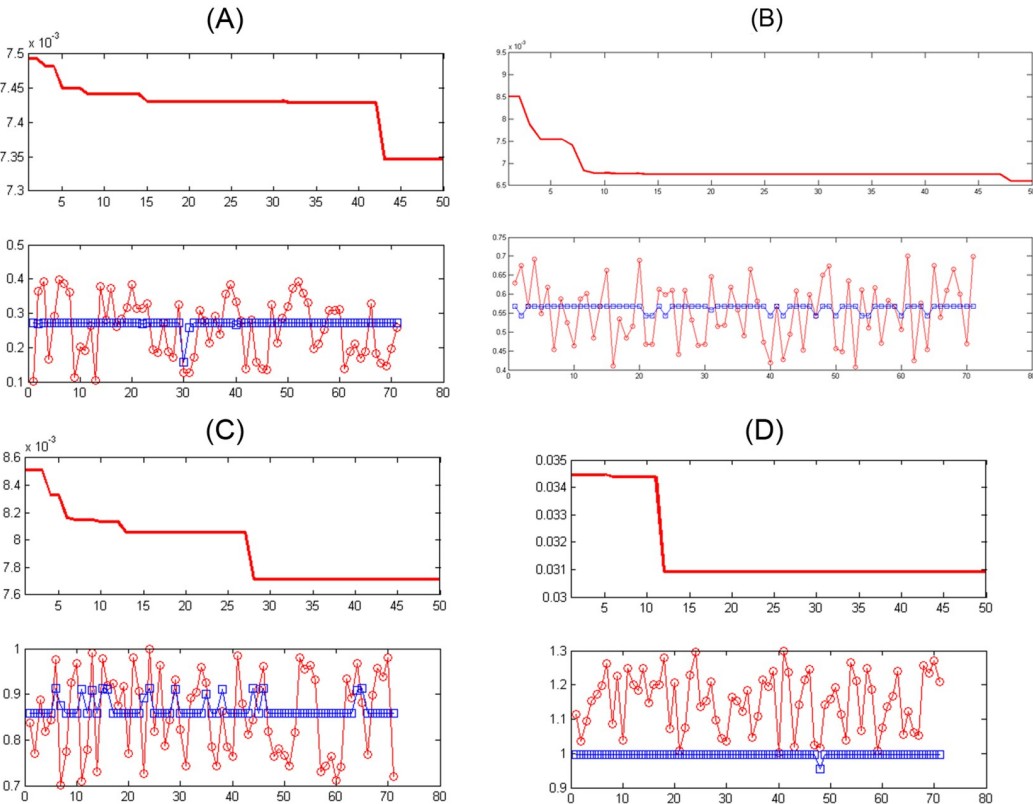

**Fig 3.** Convergence of the answers of the objective function obtained and the distance of the obtained answer (blue chart) in each generation change in ANN-BTA compared to the real answer (red chart) in training samples with ANN-BTA: (Fig 3A)[0.1, 0.4], (Fig 3B)[0.4, 0.7], (Fig 3C)[0.7, 1] and (Fig 3D)[1, 1.3] range.

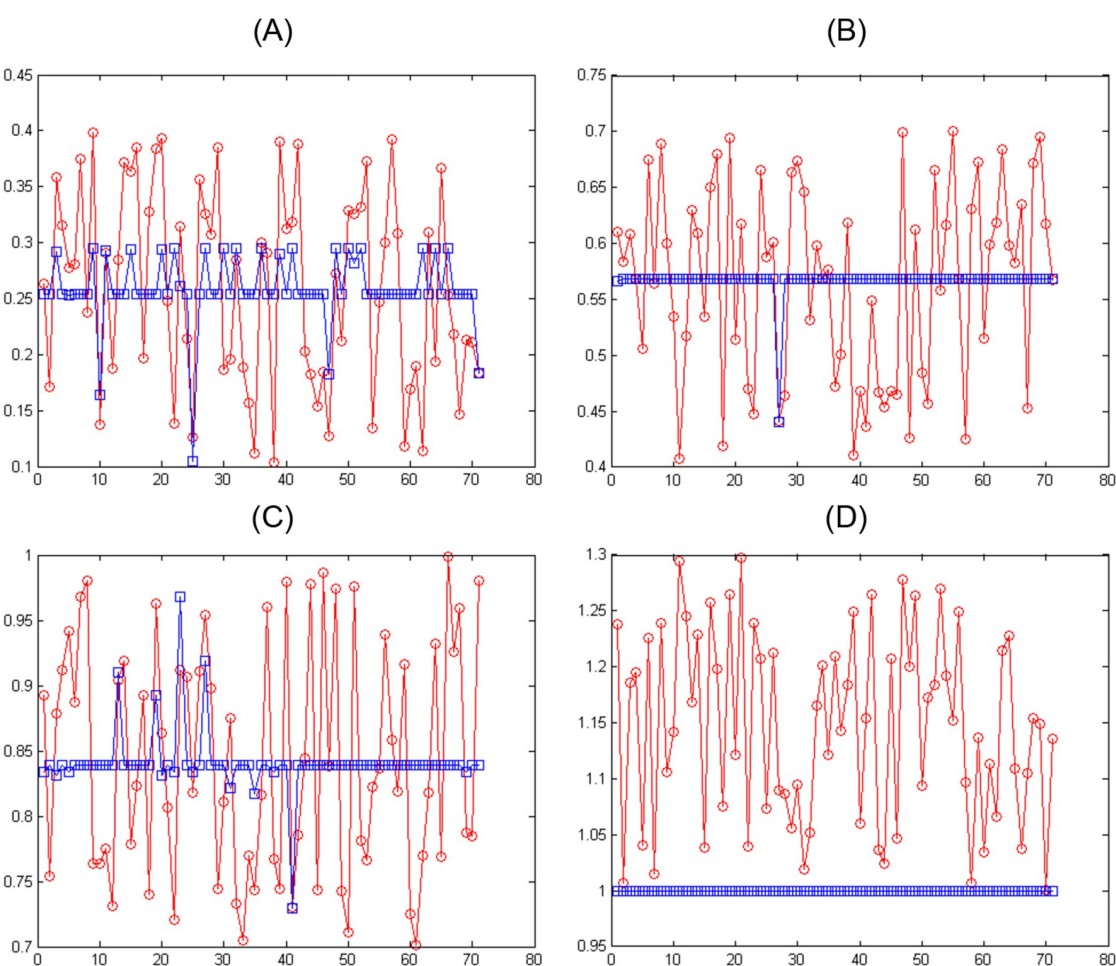

**Fig 4.** The distance of the obtained answer (blue chart) in each generation change compared to the real answer (red chart) in training samples with ANN-PSO: (Fig 4A)[0.1, 0.4], (Fig 4B)[0.4, 0.7], (Fig 4C)[0.7, 1] and (Fig 4D)[1, 1.3] range.

best run of the two ANN-BTA and ANN-PSO networks with mortality rates of [0.1, 0.4], [0.4, 0.7], [0.7, 1], [1, 1.3], [1.3, 1.7], [1.7, 2]. The results showed that considering a mortality rate above 1% does not provide an accurate approximation of the input data for people affected by COVID-19.

Figs 7–10 show the distance between the training and test samples and the regression line obtained by the two ANN-BTA and ANN-PSO networks. The results confirmed that the distance between points within the mortality rate above 1% drastically increased in both networks in the training and test phases. Looking at Figs 3 to 12, it is clear that ranges above 1% cannot provide adequate functional approximation of the input data. Therefore, in order to determine the exact ranges for estimating the death cases resulting from COVID-19, we would focus on ranges smaller than one, i.e. ranges [0.1, 0.4], [0.4, 0.7], [0.7, 1] so that we can determine the range that is the best result of the execution of two ANN-BTA and ANN-PSO networks and set that ranges as our estimation criterion for the mortality rate of cases infected by COVID-19 in the period of February 19, 2020 to May 18, 2020 in Iran. For this purpose, according to the parameters of 1 by determining random numbers in the specified three ranges, the results in terms of the total squared error in both ANN-BTA and ANN-PSO networks in 20 different

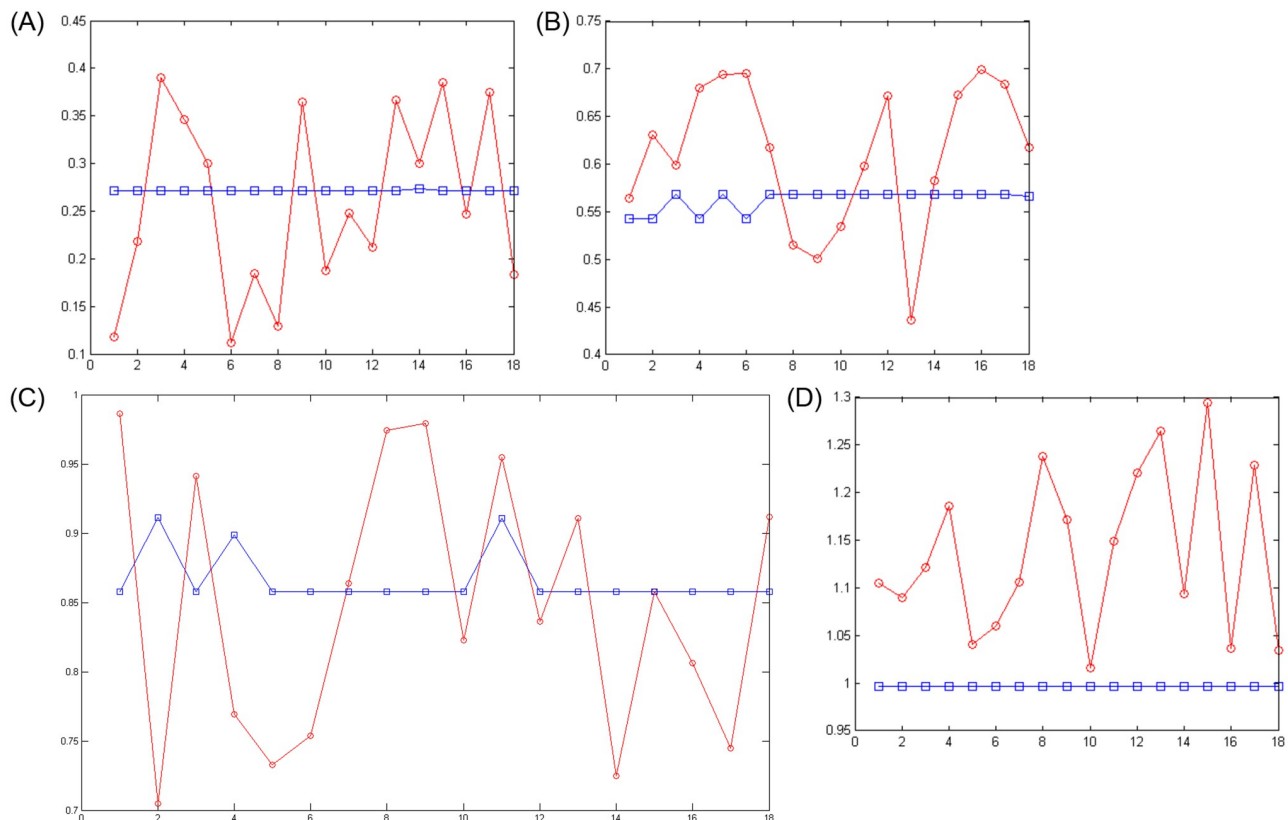

**Fig 5.** The distance of the obtained answer (blue chart) in each generation change in ANN-BTA compared to the real answer (red chart) in test samples: (Fig 5A)[0.1, 0.4], (Fig 5B)[0.4, 0.7], (Fig 5C)[0.7, 1] and (Fig 5D)[1, 1.3] range.

implementations are presented in Tables 1 and 2. Also, Tables 3 and 4 show the arithmetic and geometric mean of these 20 executions with training and geometric data. Figs 11 and 12 show the closest points to the regression line at ranges of [0.1, 0.4], [0.4, 0.7], [0.7, 1], and [1, 1.13] in the ANN-BTA network in both training and test data groups, respectively. And as it can be seen the closest point is in both training data group and test data group in the range of [0.1, 0.4] and with a value less than 0.275.

## 4.2 Estimation of the basic reproduction number and, Estimating the number of cases, mortality rate, number of deaths and time of herd immunity without effective intervention the number of COVID-19 cases in the period of February 19, 2020 to May 18, 2020 in Iran

The results clearly showed that when the mortality rate is determined above 1%, regarding COVID-19 patients, as the output of the input data, the sum of the mean squared errors increases dramatically, which confirms that basically the mortality rate and the cause of the disease should be sought in the range of [0.1]. Tables 5 and 6 show the sum of the mean squared errors in the best run of ANN -BTA and ANN-PSO networks at the mortality rates of [0.1,0.4], [0.4,0.7], [0.7,1] and [1,1.3]. Both ANN-PSO and ANN-BTA neural networks had the lowest network error in the range of [1,1.2] in estimating the basic reproduction number, that confirms that the basic reproduction number of COVID-19 patients should be searched in this

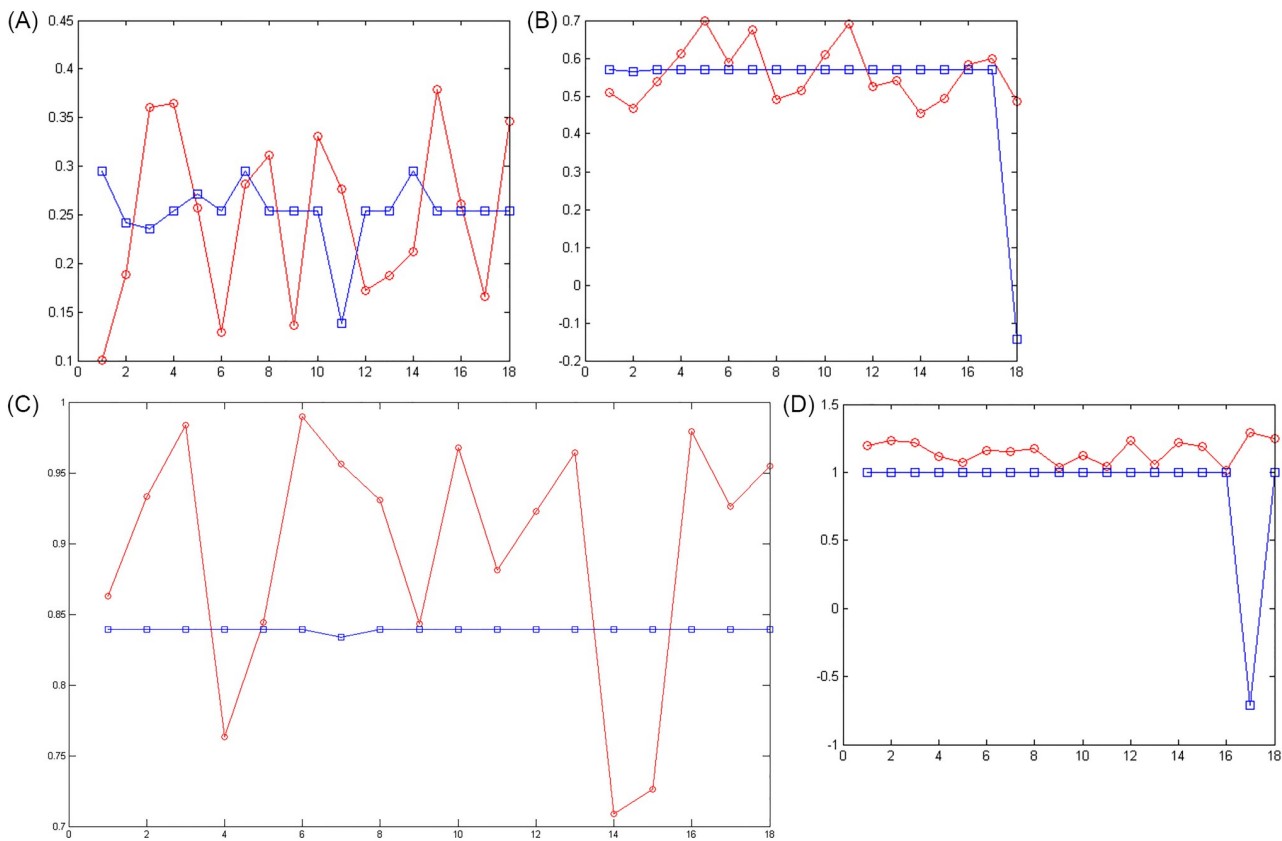

**Fig 6.** The distance of the obtained answer (blue chart) in each generation change in ANN-PSO compared to the real answer (red chart) in test samples: (Fig 6A)[0.1, 0.4], (Fig 6B)[0.4, 0.7], (Fig 6C)[0.7, 1] and (Fig 6D)[1, 1.3] range.

range. Tables 5 and 6 show the ANN-BTA and ANN-PSO error rates in the ranges of [0.8,1], [1,1.2], [1.2,1.4] and [1.4,1.6], respectively. The point to consider is when the basic reproduction number is estimated higher than 1.4, in which the network error is greatly increased. Comparing the two tables, we find out that the proposed designed network's errors (ANN-BTA) are less in all ranges. This goes back to neural network design. Because part of the network error is related to the learning algorithm that the neural network benefits from. Considering the reduction of error in all the evaluated ranges, it can be concluded that ANN-BTA has learned better and as a result has obtained better results in network tests.

For more accurate estimation of basic reproduction number and based on the proposed method, the interval [1,1.2] was limited and more limited, and after several steps of interval limitation based on the minimum value of MSE, the network reached its lowest error at 1.45 (Table 7). This value would be called the basic reproduction number and the value obtained by ANN-BTA network. The minimum value of MSE for estimating basic reproduction number in ANN-PSO network was 1.65 (Table 8). In both ANN-BTA and ANN-PSO networks, these intervals can be narrowed down to achieve lower MSE values.

Regarding the estimation of mortality rate among COVID-19 patients, it is interesting that both ANN-PSO and ANN-BTA neural networks had the lowest network error in the range of [0.1,0.4] and this confirms that the mortality rate among those with COVID-19 should be sought in this area. Tables 9 and 10 show the ANN-BTA and ANN-PSO error rates in the

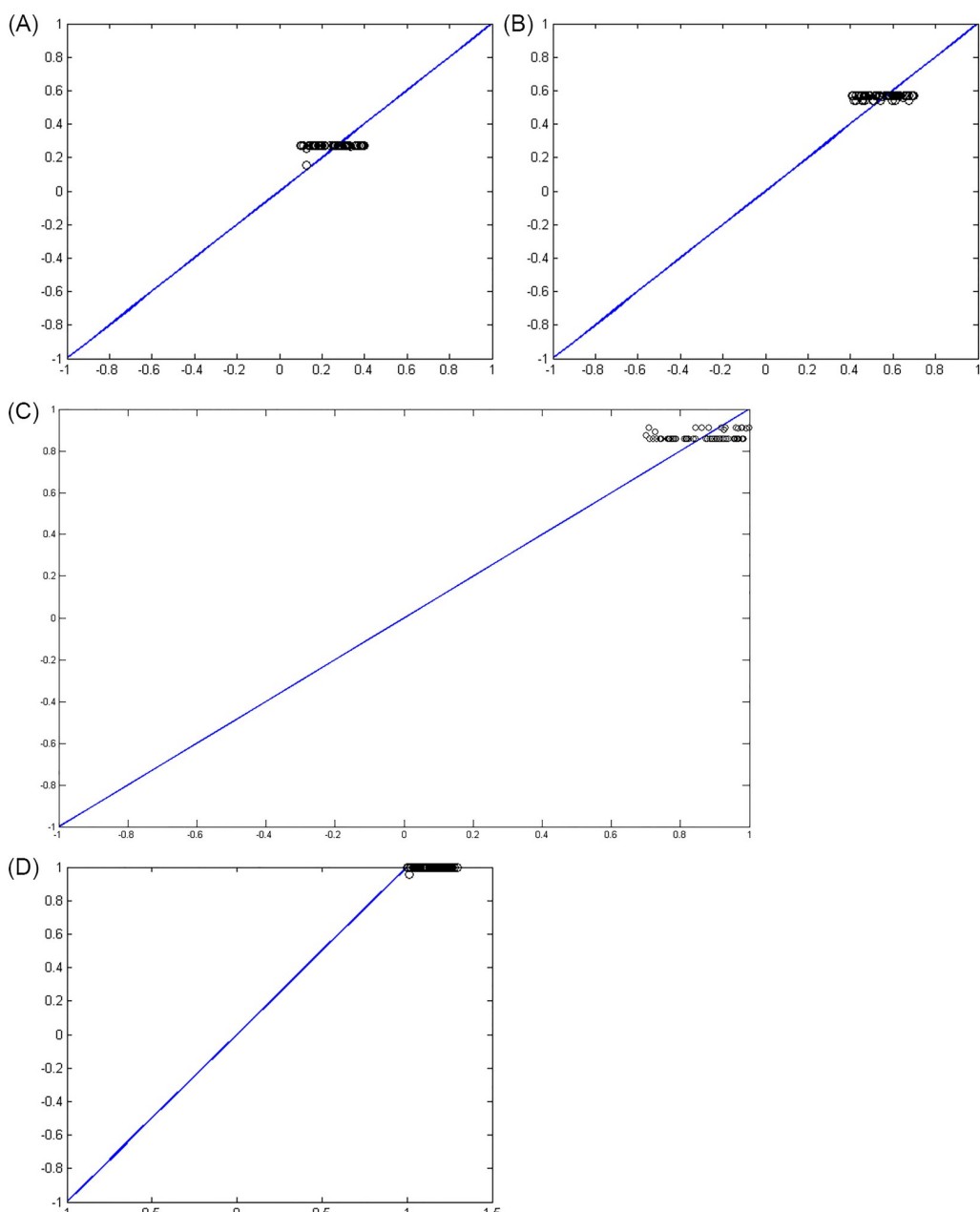

**Fig 7.** The distance between the real answers and the regression line (regression function) obtained with ANN-BTA in the training samples: (Fig 7A)[0.1, 0.4], (Fig 7B)[0.4, 0.7], (Fig 7C)[0.7, 1] and (Fig 7D)[1, 1.3] range.

ranges of [0.1,0.4], [0.4,0.7], [0.7,1] and [1,1.3], respectively. The point to consider is when the mortality rate is estimated to be 1, in which case the network error increases significantly.

In order to more accurately estimate the growth rate based on the proposed method, the interval [0.1,0.4] was limited and narrowed, and after a few steps of interval limitation based on the minimum value of MSE, the network reached its lowest error rate of 0.275 (Table 11). This value can be called the basic reproduction number value obtained by ANN-BTA network. The minimum value of MSE for estimating mortality rate in ANN-PSO network was also

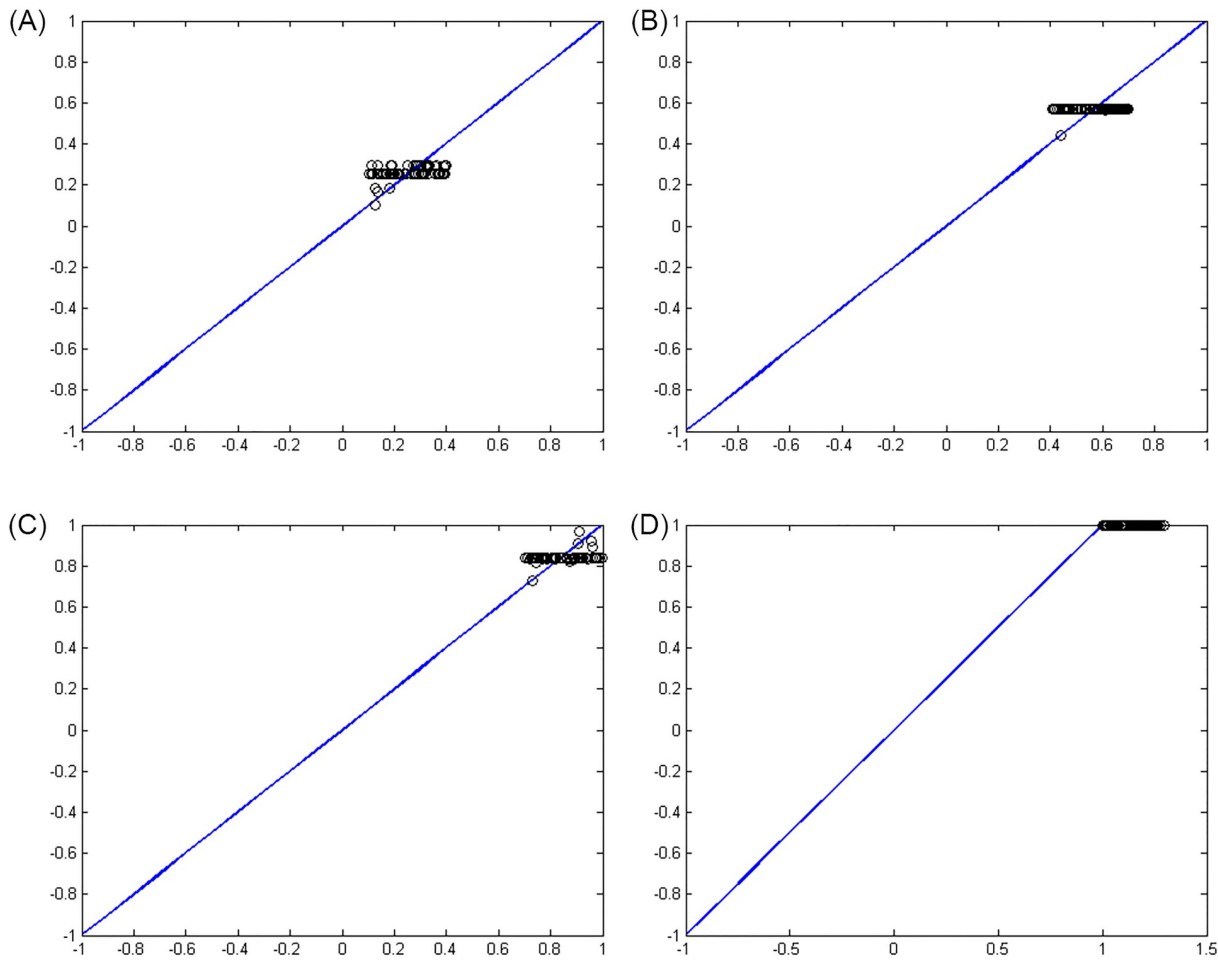

**Fig 8.** The distance between the real answers and the regression line (regression function) obtained with ANN-PSO in the training samples: (Fig 8A)[0.1, 0.4], (Fig 8B)[0.4, 0.7], (Fig 8C)[0.7, 1] and (Fig 8D)[1, 1.3] range.

obtained at 0.275 (Table 12). In both ANN-BTA and ANN-PSO networks, these intervals can be narrowed to achieve lower MSE values.

Table 13 estimated number of patients with COVID-19 based on the mortality rate of patients obtained by ANN-BTA network in Table 11 (ie 0.275) and the mortality declared by the Ministry of Health of Iranduring the time period of February 19, 2020 to May 18, 2020 in Iran. Also, based on this estimation, the basic reproduction number is calculated daily.

At the bottom of the Table 13 the arithmetic mean of the basic reproduction number in 89 days is shown. It can be clearly seen that the basic reproduction number obtained by ANN-BTA (i.e. 1.045) mentioned in tbl1000 is very close to the arithmetic mean of basic reproduction number and in 89 days (i.e. 1.0411).

The model proposed in the present study showed that if sever quarantine restrictions are not applied and Iranian government does not impose effective interventions, about 60% to 70% of the population (it means around 49 to 58 million people) would be afflicted by COVID-19 during June to September 2021. Therefore, it could be predicted when the herd immunity may take place in Iran. The proposed model contributes to the hypothesis that possibly 232000 death cases might be recorded before reaching herd immunity.

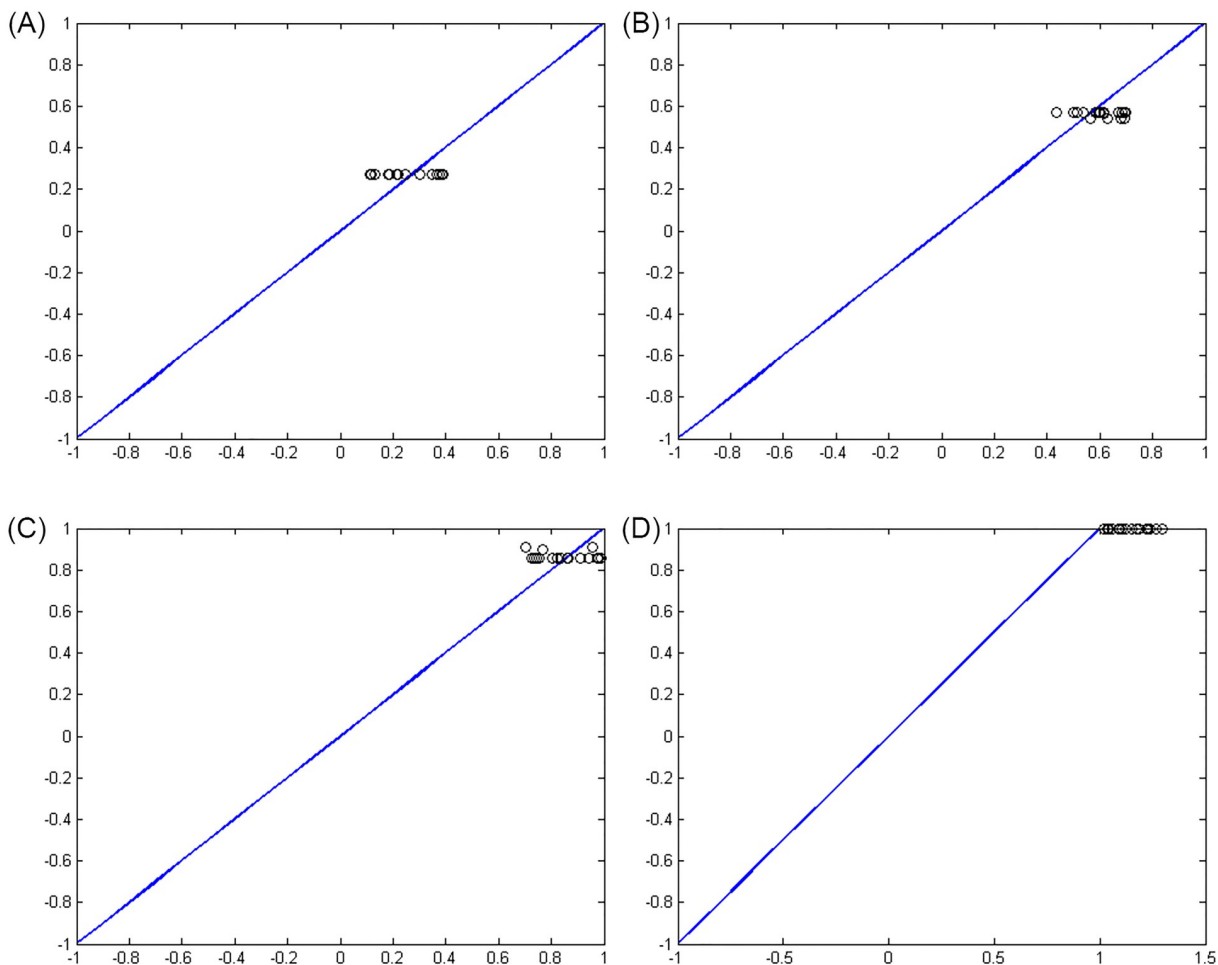

**Fig 9.** The distance between the real answers and the regression line (regression function) obtained with ANN-BTA in the test samples: (Fig 9A)[0.1, 0.4], (Fig 9B)[0.4, 0.7], (Fig 9C)[0.7, 1] and (Fig 9D)[1, 1.3] range.

### 4.3 Effects of interventions and definition of quarantine levels as well as weather variant on mortality rate of cases with COVID-19 in the period of February 19, 2020 to May 18, 2020 in Iran, obtained from ANN-BTA network

In order to evaluate the role of government's interventions for quarantine levels in Iran as well as the weather variant in two different implementations, by deleting the data column of each factor we see the effects of the deletion on the error rate of the network. It is crystal clear that if the removal of each column causes a significant increase in error, in error, it will have far more effects than other factors on the mortality rate and the incidence of COVID-19 virus in the period defined in Iran. Therefore, considering that in the previous two sections the fitness of ANN-BTA network compared to ANN-PSO network was proven, in this section we first implemented the network without the government's intervention order factor having in mind the mortality rate output of [0.1, 0.4] and evaluated its effects on the error value. Then, by removing the weather variant factor with the pre-mentioned outputs we performed this assessment and evaluation. Table 14 shows the effect of the lack of a governmental intervention

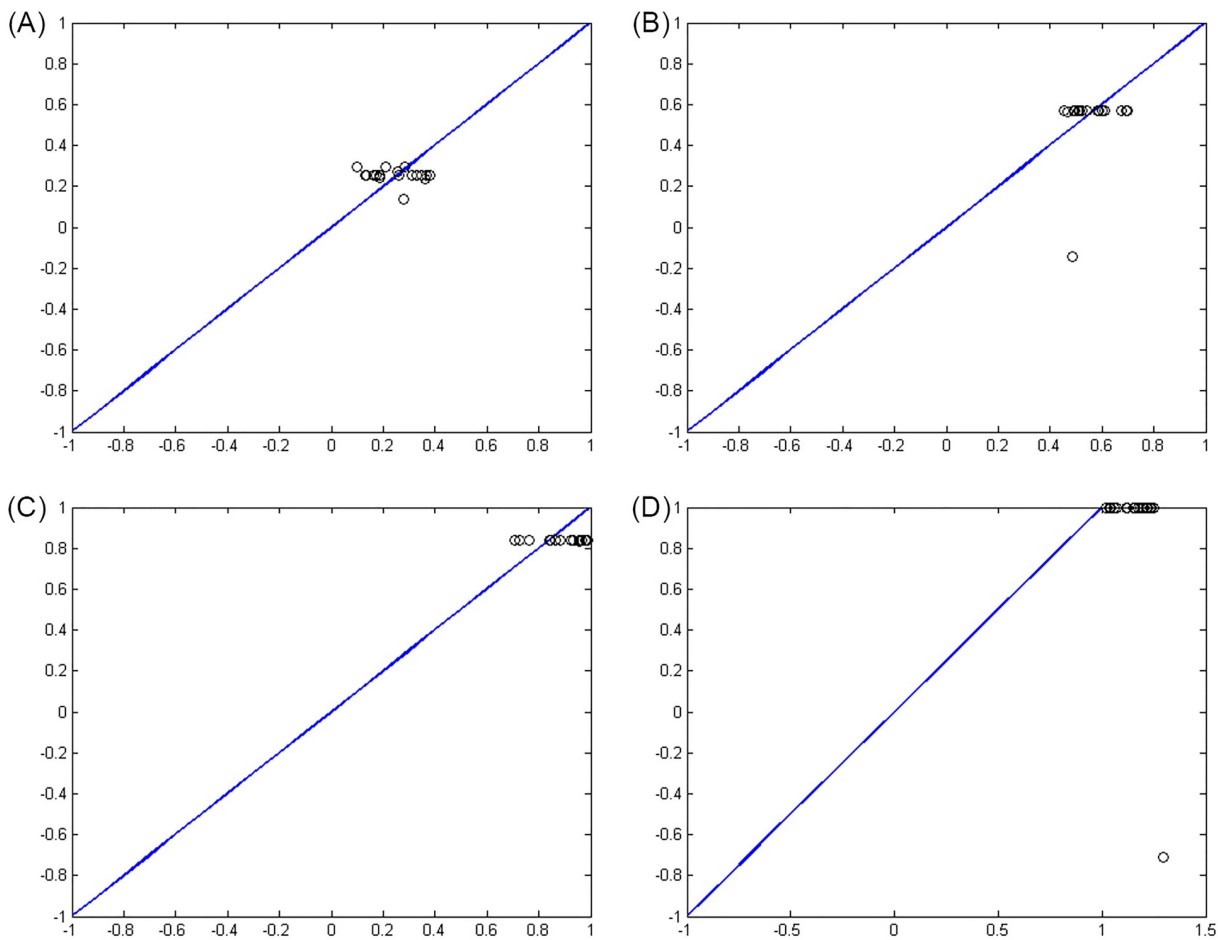

**Fig 10.** The distance between the real answers and the regression line (regression function) obtained with ANN-PSO in the test samples: (Fig 10A)[0.1, 0.4], (Fig 10B)[0.4, 0.7], (Fig 10C)[0.7, 1] and (Fig 10D)[1, 1.3] range.

order factor on the total squared error of ANN-PSO network with the mortality rate output of [0.1, 0.4]. Table 11 shows the effect of the lack of weather factor on the total squared error of ANN-PSO network of the government with the mortality rate output of [0.1, 0.4]. The results showed that the role of government's interventions order on network error changes is far greater than that of weather variant. Of course, this role is very insignificant. For instance, in the range of [0.1, 0.4] in the training data this effect on error is only 0.0001.Therefore, it can be said that the proposed model of the present study estimates the role of weather variant in the mortality rate ineffective and the role of quarantine policies implemented by the Iran government in the mortality rate in cases with COVID-19 very little and insignificant in the period of February 19, 2020 to May 18, 2020 in Iran. Table 15 shows the superiority of the present study's method in terms of processing with accurate data, the possibility of processing with low data volume and noisy data, and the process of epidemiological knowledge discovery. The present method not only does not require accurate data and has the ability to go through the process of predicting epidemiological processes with accessible data, but also the process of discovering epidemiological knowledge in the present study does not require expert knowledge and with any level of knowledge the epidemiological knowledge can be understood.

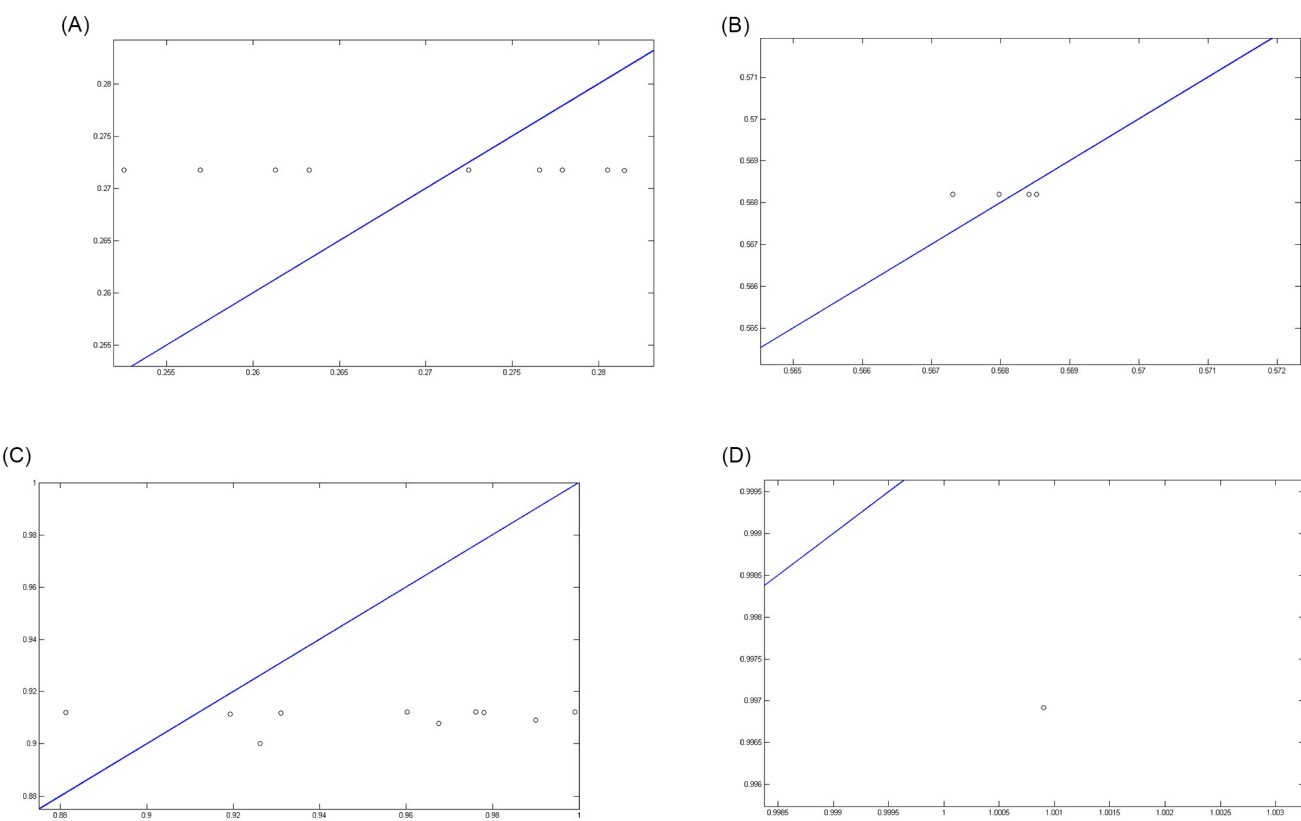

**Fig 11.** The closest answer to the regression line (regression function) obtained with ANN-BTA in the training samples: (Fig 11A)[0.1, 0.4], (Fig 11B)[0.4, 0.7], (Fig 11C)[0.7, 1] and (Fig 11D)[1, 1.3] range.

## 5 Discussion

In the present study, based on data obtained according to daily deaths, daily mortality rates, government interventions in order to apply quarantine policies, as well as the average age range of the country and daily temperature differences in the period from February 19, 2020 to May 18, 2020, an epidemiological model was proposed according to two neural networks each with a specific learning algorithm to estimate mortality rate and the rate of growth of the COVID-19 virus; and also the effects of government's intervention order on quarantine and weather variant conditions, and herd immunity. These two networks were ANN-BTA and ANN-PSO Networks. ANN-BTA network showed a much lower error rate than ANN-PSO network. Therefore, by focusing on this network the following results were obtained from its various executions.The importance of the improvements that have taken place in the results lies in the fact that these results are obtained with data that the accuracy of all this data is not verifiable and in other words we do not encounter reliable data. Therefore, slight improvements in the results of the present study with data obtained in the early COVID-19 pandemic period promise that when reliable data are not available, we can rely on the learning method of the present study, which over time due to learning, the system's answers are more accurately and complete. On the other hand, the improved obtained answers are not necessarily obtained with the help of an expert epidemiologist, and with any level of epidemiological knowledge, knowledge can be extracted. Another problem that adds to the value of the responses obtained is that by processing large volumes of data that are definitely noisy, a response worthy of

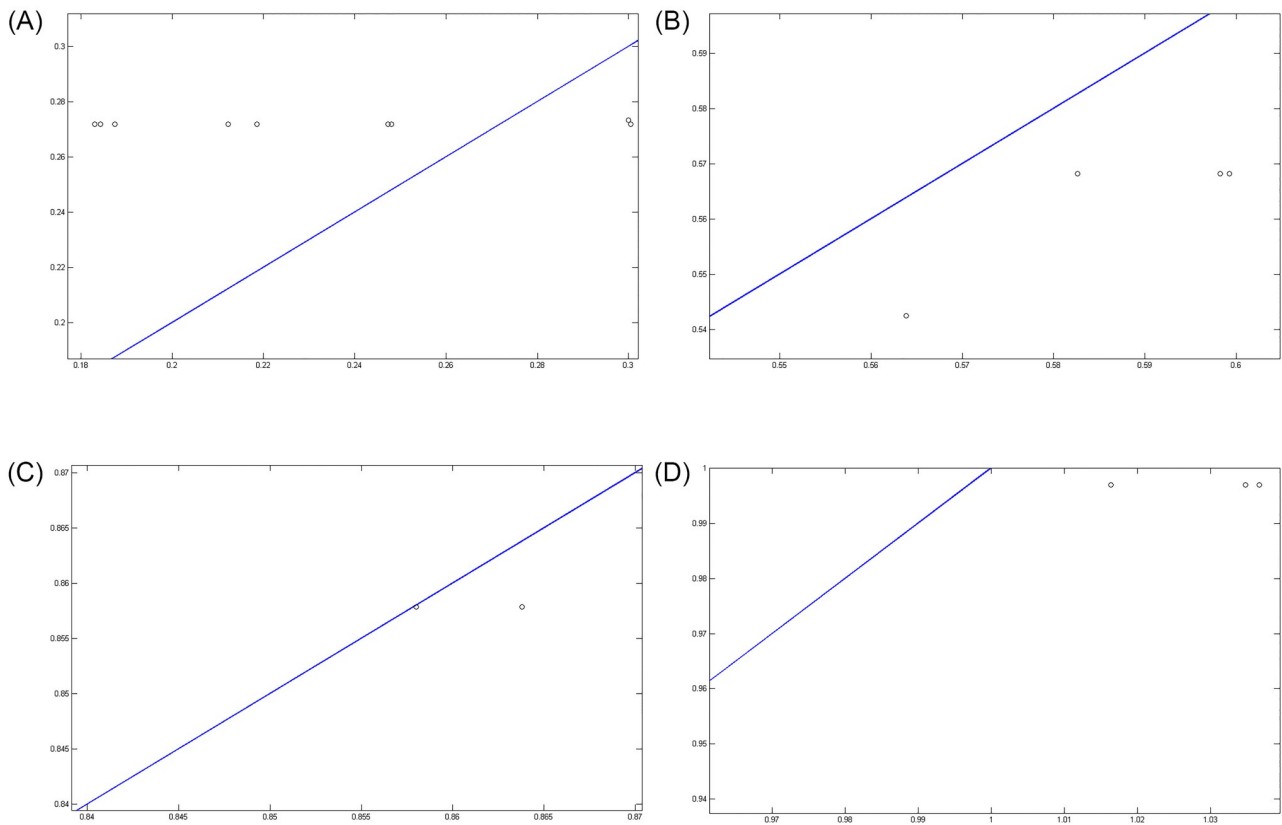

**Fig 12.** The closest answer to the regression line (regression function) obtained with ANN-BTA in the testing samples: (Fig 12A)[0.1, 0.4], (Fig 12B) [0.4, 0.7], (Fig 12C)[0.7, 1] and (Fig 12D)[1, 1.3] range.

attention with little improvement over previous methods is obtained. As such, COVID-19 growth rate in the period of February 19, 2020 to May 18, 2020 in Iran is illustrated in Fig 13.

1. Over the course of 90 days, the 45th day after the first COVID-19 death was reported; the highest death toll was reported to be 158. Also, the highest virus growth rate is related to the eighth and eighteenth days after the report of the first death due to COVID-19. The growth rate of these days is 2.33. The geometric mean of the growth rate for the whole 90 days is 1.04.

2. The results clearly show that when the rate is determined above 1% as the mortality rate from COVID-19 for input data as output, the total mean squared errors increases dramatically, which confirms that the mortality rate from this disease basically should be sought in range [0, 1]. ANN-BTA Network on point 0.275 provided the best results in terms of the total training and test squared errors of network as the mortality rate.

3. Considering that the mortality rate obtained in the mentioned period is in the range of [0.1, 0.4], therefore, by placing this mortality rate for each number of deaths per day, an estimate can be made based on estimating the average mortality rate for all values between [0.1, 0.4]. That is, in the present study, we calculated the rate of infection and the number of cases for the obtained values of 0.1, 0.2, 0.3, and 0.4, as well as the closest value to the regression line

**Table 1. The results of the sum of the squared error in 20 implementation of ANN-BTA network in three ranges of [0.1, 0.4], [0.4, 0.7], [0.7, 1].**

| Type of Error → | MSEts | MSEtr | MSEts | MSEtr | MSEts | MSEtr |
|---|---|---|---|---|---|---|
| Iteration ↓ | [0.7,1] | | [0.4,0.7] | | [0.1,0.4] | |
| 1 | 0.0114 | 0.0073 | 0.0102 | 0.0079 | 0.0078 | 0.0073 |
| 2 | 0.0124 | 0.0074 | 0.0092 | 0.0081 | 0.0074 | 0.0078 |
| 3 | 0.0090 | 0.0075 | 0.0083 | 0.0083 | 0.0093 | 0.0075 |
| 4 | 0.0095 | 0.0080 | 0.0103 | 0.0069 | 0.0058 | 0.0083 |
| 5 | 0.0080 | 0.0084 | 0.0081 | 0.0083 | 0.0070 | 0.0080 |
| 6 | 0.0101 | 0.0073 | 0.0077 | 0.0085 | 0.0067 | 0.0079 |
| 7 | 0.0080 | 0.0080 | 0.0051 | 0.0090 | 0.0078 | 0.0075 |
| 8 | 0.0107 | 0.0077 | 0.0080 | 0.0085 | 0.0068 | 0.0081 |
| 9 | 0.0081 | 0.0081 | 0.0110 | 0.0081 | 0.0089 | 0.0079 |
| 10 | 0.0103 | 0.0080 | 0.0106 | 0.0077 | 0.0100 | 0.0071 |
| 11 | 0.0058 | 0.0082 | 0.0067 | 0.0086 | 0.0062 | 0.0080 |
| 12 | 0.0062 | 0.0083 | 0.0096 | 0.0080 | 0.0049 | 0.0082 |
| 13 | 0.0096 | 0.0079 | 0.0066 | 0.0087 | 0.0095 | 0.0071 |
| 14 | 0.0098 | 0.0077 | 0.0100 | 0.0079 | 0.0125 | 0.0066 |
| 15 | 0.0063 | 0.0085 | 0.0100 | 0.0081 | 0.0070 | 0.0077 |
| 16 | 0.0107 | 0.0080 | 0.0103 | 0.0080 | 0.0083 | 0.0078 |
| 17 | 0.0078 | 0.0080 | 0.0131 | 0.0074 | 0.0078 | 0.0078 |
| 18 | 0.0086 | 0.0080 | 0.0142 | 0.0070 | 0.0086 | 0.0076 |
| 19 | 0.0068 | 0.0082 | 0.0076 | 0.0086 | 0.0075 | 0.0080 |
| 20 | 0.0077 | 0.0080 | 0.0100 | 0.0079 | 0.0081 | 0.0077 |

**Table 2. The results of the sum of the squared error in 20 implementation of ANN-PSO network in three ranges of [0.1, 0.4], [0.4, 0.7], [0.7, 1].**

| Type of Error → | MSEts | MSEtr | MSEts | MSEtr | MSEts | MSEtr |
|---|---|---|---|---|---|---|
| Iteration ↓ | [0.7,1] | | [0.4,0.7] | | [0.1,0.4] | |
| 1 | 0.377 | 0.0063 | 0.0039 | 0.0091 | 0.0089 | 0.0086 |
| 2 | 0.0049 | 0.0077 | 0.0092 | 0.0073 | 0.0096 | 0.0084 |
| 3 | 0.0144 | 0.0062 | 0.0072 | 0.0076 | 0.0078 | 0.0086 |
| 4 | 0.0073 | 0.0066 | 0.0091 | 0.0070 | 0.0080 | 0.0085 |
| 5 | 0.0073 | 0.0062 | 0.0084 | 0.0072 | 0.0093 | 0.0087 |
| 6 | 0.0046 | 0.0077 | 0.0071 | 0.0083 | 0.0087 | 0.0092 |
| 7 | 0.0092 | 0.0063 | 0.0073 | 0.0082 | 0.0319 | 0.0085 |
| 8 | 0.0063 | 0.0072 | 0.0094 | 0.0076 | 0.0088 | 0.0089 |
| 9 | 0.0082 | 0.0075 | 0.0075 | 0.0080 | 0.0118 | 0.0078 |
| 10 | 0.0114 | 0.0061 | 0.0119 | 0.0072 | 0.0077 | 0.0089 |
| 11 | 0.0065 | 0.0077 | 0.0059 | 0.0081 | 0.0113 | 0.0084 |
| 12 | 0.1140 | 0.0074 | 0.0094 | 0.0077 | 0.0118 | 0.0081 |
| 13 | 0.0075 | 0.0066 | 0.0107 | 0.0074 | 0.0106 | 0.0092 |
| 14 | 0.0084 | 0.0073 | 0.0145 | 0.0069 | 0.0116 | 0.0077 |
| 15 | 0.0086 | 0.0072 | 0.0098 | 0.0073 | 0.0097 | 0.0085 |
| 16 | 0.0109 | 0.0069 | 0.0162 | 0.0067 | 0.0163 | 0.0082 |
| 17 | 0.0073 | 0.0073 | 0.0101 | 0.0073 | 0.0101 | 0.0080 |
| 18 | 0.0070 | 0.0076 | 0.0089 | 0.0076 | 0.0078 | 0.0092 |
| 19 | 0.0140 | 0.0059 | 0.0069 | 0.0081 | 0.0126 | 0.0079 |
| 20 | 0.0084 | 0.0073 | 0.0099 | 0.0075 | 0.0078 | 0.0092 |

**Table 3. Arithmetic and geometric mean obtained from the sum of squared error in 20 implementations of ANN-BTA network in three ranges of [0.1, 0.4], [0.4, 0.7], [0.7, 1].**

| Average type ↓/ Squared Error Range → | [0.7,1] | [0.4,0.7] | [0.1,0.4] |
|---|---|---|---|
| Arithmetic mean tr | 0.0079 | 0.0081 | 0.0077 |
| Geometric mean tr | 0.0079 | 0.0081 | 0.0077 |
| Arithmetic mean ts | 0.0088 | 0.0095 | 0.0079 |
| Geometric mean ts | 0.0087 | 0.0092 | 0.0077 |

**Table 4. Arithmetic and geometric mean obtained from the sum of the squared error in 20 implementation of ANN-PSO network in three ranges of [0.1, 0.4], [0.4, 0.7], [0.7, 1].**

| Average type ↓/ Squared Error Range → | [0.7,1] | [0.4,0.7] | [0.1,0.4] |
|---|---|---|---|
| Arithmetic mean tr | 0.0070 | 0.0076 | 0.0085 |
| Geometric mean tr | 0.0069 | 0.0076 | 0.0085 |
| Arithmetic mean ts | 0.0152 | 0.0092 | 0.0103 |
| Geometric mean ts | 0.0100 | 0.0088 | 0.0101 |

**Table 5. Total mean square error (MSE) of training and experimental data in the best implementation of ANN-BTA network for estimating basic reproduction number related to COVID-19 patients in the ranges of [0.8,1], [1,1.2], [1.2,1.4] And [1.4,1.6].**

| Estimated basic reproduction numberand | MSE Train | MSE Test |
|---|---|---|
| [0.8, 1] | 0.0068 | 0.0071 |
| **[1,1.2]** | **0.0058** | **0.0061** |
| [1.2, 1.4] | 0.1466 | 0.0182 |
| [1.4, 1.6] | 0.1171 | 0.1335 |

**Table 6. Total mean square error (MSE) of training and experimental data in the best implementation of ANN-BTA network for estimating basic reproduction number related to COVID-19 patients in the ranges of [0.8,1], [1,1.2], [1.2,1.4] And [1.4,1.6].**

| Estimated basic reproduction numberand | MSE Train | MSE Test |
|---|---|---|
| [0.8, 1] | 0.0071 | 0.0073 |
| **[1,1.2]** | **0.0060** | **0.0066** |
| [1.2, 1.4] | 0.0194 | 0.0187 |
| [1.4, 1.6] | 0.1201 | 0.136 |

**Table 7. The MSE value of ANN-BTA network after interval limiting [1,1.2] and obtaining the lowest error rate in estimating basic reproduction numberand.**

| Estimated basic reproduction numberand | MSE Train | MSE Test |
|---|---|---|
| 1.040 | 0.0053 | 0.0056 |
| **1.045** | **0.0049** | **0.0052** |
| 1.050 | 0.0062 | 0.0070 |
| 1.055 | 0.0068 | 0.0072 |

**Table 8. The MSE value of ANN-PSO network after interval limiting [1,1.2] and obtaining the lowest error rate in estimating basic reproduction numberand.**

| Estimated basic reproduction numberand | MSE Train | MSE Test |
|---|---|---|
| 1.060 | 0.0061 | 0.0064 |
| **1.065** | **0.0055** | **0.0059** |
| 1.070 | 0.0069 | 0.0074 |
| 1.075 | 0.0076 | 0.0081 |

**Table 9. Total mean square error (MSE) of training and testing data in the best implementation of the ANN-BTA network for estimating the mortality rate of COVID-19 patients in the ranges of [0.1,0.4], [0.4,0.7], [0.7, 1], [1.1.3], [1.3,1.7] and [1.7,2].**

| Estimated mortality rate range | MSE Train | MSE Test |
|---|---|---|
| [0.1,0.4] | **0.0061** | **0.0052** |
| [0.4, 0.7] | 0.0081 | 0.0090 |
| [0.7, 1] | 0.0066 | 0.0082 |
| [1, 1.3] | 0.0301 | 0.0336 |
| [1.3, 1.7] | 0.2546 | 0.2584 |
| [1.7, 2] | 0.7425 | 0.7174 |

obtained, i.e. 0.275. The sum of the cases affected by COVID-19 for this growth rate is 2566200 cases.

4. The proposed model of the present study estimates the role of weather variant in the rate of infection ineffective in the mortality rate and also it estimates the role of quarantine policies implemented by the Iranian government at mortality rate of people infected by COVID-19

**Table 10. Total mean square error (MSE) of training and testing data in the best implementation of the ANN-PSO network for estimating the mortality rate of COVID-19 patients in the ranges of [0.1,0.4], [0.4,0.7], [0.7, 1], [1.1.3], [1.3,1.7] and [1.7,2].**

| Estimated mortality rate range | MSE Train | MSE Test |
|---|---|---|
| [0.1,0.4] | **0.0086** | **0.0078** |
| [0.4, 0.7] | 0.0072 | 0.0084 |
| [0.7, 1] | 0.0062 | 0.0073 |
| [1, 1.3] | 0.0336 | 0.0366 |
| [1.3, 1.7] | 0.2616 | 0.2594 |
| [1.7, 2] | 0.7176 | 0.6967 |

**Table 11. The MSE value of ANN-BTA network after interval limiting [1,1.2] and obtaining the lowest error rate in estimating mortality rate.**

| Estimated mortality rate | MSE Train | MSE Test |
|---|---|---|
| 0.270 | 0.0064 | 0.0066 |
| **0.275** | **0.0051** | **0.0054** |
| 0.280 | 0.0058 | 0.0061 |
| 0.285 | 0.0082 | 0.0083 |

**Table 12. The MSE value of ANN-PSO network after interval limiting [1,1.2] and obtaining the lowest error rate in estimating mortality rate.**

| Estimated mortality rate | MSE Train | MSE Test |
|---|---|---|
| 0.270 | 0.0072 | 0.0080 |
| **0.275** | **0.0069** | **0.0074** |
| 0.280 | 0.0078 | 0.0088 |
| 0.285 | 0.0106 | 0.0113 |

very low in the period of February 19, 2020 to May 18, 2020 in Iran. For example, in the range of [0.1, 0.4] in training data, the impact of this amount of error is only 0.0001.

5. Failure to apply the average age factor of Iranian population leads to significant errors in training and test samples. Therefore, the method presented in the present study confirms the role of average age of the country in the mortality rate of cases with COVID-19 in the period of February 19, 2020 to May 18, 2020 in Iran.

## 6 Conclusion

Based on the results of the present study, weather variant change and quarantine policies have played no role in the mortality rate of COVID-19 virus disease. But the findings of the present study show that the age factor determines the mortality rate caused by the COVID-19 virus. According to the proposed model it is predicted that the herd immunity in Iran would take place in June to September 2021 if sever quarantine restrictions are not applied and Iranian government does not impose effective interventions, possibly 232000 death cases from COVID-19 might be recorded before reaching herd immunity. Although, at first glance the findings of the study may be obvious and to some extent approved-due to the information that has been obtained since the beginning of the pandemic of COVID-19 virus- the mortality rate, basic reproduction number and/or prediction of the time of herd immunity, and the effectiveness of quarantine policies in a country like Iran could be elaborated on. However, the importance of the findings of the present study come forth when we are aware that they were obtained through the available data such as meteorological or fuzzy ones with much noise and errors on how to apply quarantine policies in Iran; second, the range of these data was so limited. Thus, the present study shows that the epidemiological model based on the proposed machine learning algorithm (with little available noisy data) could be used, and perhaps persuading health policymakers to be more strict over quarantine policies for further executions in future.

On the other hand, creative works such as combining artificial neural network with human intelligence algorithm based on BTA (bus transportation algorithm), to be used in an epidemiological model based on the current study's machine learning method called ANN-BTA network, showed more efficiency and performed better compared to the other advanced methods such as ANN-PSO in proposed epidemiological model; numerous tests approved the above mentioned issue.

The results of the present study showed that, compared with machine learning based method, due to limitations in defining their parameters as well as limitations in data type coverage, the conventional methods are not efficient in calculating epidemiological indices such as productivity rate index, mortality rate, predicting the number of cases and deaths, and even

**Table 13. Estimation of basic reproduction number and number of patients per day in the period of February 19, 2020 to May 18, 2020 in Iran based on the mortality rate obtained by the ANN-BTA network and the mortality rate declared by the Ministry of Health of Iran.**

| Day | Number of deaths according to the Ministry of Health of Iran | Estimation of the number of cases based on the mortality rate obtained by the ANN-BTA network(0.275) | The daily basic reproduction number |
|---|---|---|---|
| 1 | 2 | 727.27 | 0 |
| 2 | 2 | 727.27 | 1 |
| 3 | 2 | 727.27 | 1 |
| 4 | 2 | 727.27 | 1 |
| 5 | 4 | 1454.5 | 2 |
| 6 | 4 | 1454.5 | 1 |
| 7 | 3 | 1090.9 | 0.75 |
| 8 | 7 | 2545.5 | 2.3333 |
| 9 | 8 | 2909.1 | 1.1429 |
| 10 | 9 | 3272.7 | 1.125 |
| 11 | 11 | 4000 | 1.2222 |
| 12 | 12 | 4363.6 | 1.0909 |
| 13 | 11 | 4000 | 0.91667 |
| 14 | 15 | 5454.5 | 1.3636 |
| 15 | 16 | 5818.2 | 1.0667 |
| 16 | 16 | 5818.2 | 1 |
| 17 | 21 | 7636.4 | 1.3125 |
| 18 | 49 | 17818 | 2.3333 |
| 19 | 43 | 15636 | 0.87755 |
| 20 | 54 | 19636 | 1.2558 |
| 21 | 63 | 22909 | 1.1667 |
| 22 | 75 | 27273 | 1.1905 |
| 23 | 85 | 30909 | 1.1333 |
| 24 | 97 | 35273 | 1.1412 |
| 25 | 113 | 41091 | 1.1649 |
| 26 | 129 | 46909 | 1.1416 |
| 27 | 135 | 49091 | 1.0465 |
| 28 | 147 | 53455 | 1.0889 |
| 29 | 149 | 54182 | 1.0136 |
| 30 | 149 | 54182 | 1 |
| 31 | 123 | 44727 | 0.8255 |
| 32 | 129 | 46909 | 1.0488 |
| 33 | 127 | 46182 | 0.9845 |
| 34 | 122 | 44364 | 0.96063 |
| 35 | 143 | 52000 | 1.1721 |
| 36 | 157 | 57091 | 1.0979 |
| 37 | 144 | 52364 | 0.9172 |
| 38 | 139 | 50545 | 0.96528 |
| 39 | 123 | 44727 | 0.88489 |
| 40 | 117 | 42545 | 0.95122 |
| 41 | 141 | 51273 | 1.2051 |
| 42 | 138 | 50182 | 0.97872 |
| 43 | 124 | 45091 | 0.89855 |
| 44 | 134 | 48727 | 1.0806 |
| 45 | 158 | 57455 | 1.1791 |
| 46 | 151 | 54909 | 0.9557 |
| 47 | 136 | 49455 | 0.90066 |

*(Continued)*

**Table 13.** (Continued)

| Day | Number of deaths according to the Ministry of Health of Iran | Estimation of the number of cases based on the mortality rate obtained by the ANN-BTA network(0.275) | The daily basic reproduction number |
|---|---|---|---|
| 48 | 133 | 48364 | 0.97794 |
| 49 | 121 | 44000 | 0.90977 |
| 50 | 117 | 42545 | 0.96694 |
| 51 | 122 | 44364 | 1.0427 |
| 52 | 125 | 45455 | 1.0246 |
| 53 | 117 | 42545 | 0.936 |
| 54 | 111 | 40364 | 0.94872 |
| 55 | 98 | 35636 | 0.88288 |
| 56 | 94 | 34182 | 0.95918 |
| 57 | 92 | 33455 | 0.97872 |
| 58 | 89 | 32364 | 0.96739 |
| 59 | 73 | 26545 | 0.82022 |
| 60 | 87 | 31636 | 1.1918 |
| 61 | 91 | 33091 | 1.046 |
| 62 | 88 | 32000 | 0.96703 |
| 63 | 94 | 34182 | 1.0682 |
| 64 | 90 | 32727 | 0.95745 |
| 65 | 93 | 33818 | 1.0333 |
| 66 | 76 | 27636 | 0.8172 |
| 67 | 60 | 21818 | 0.78947 |
| 68 | 96 | 34909 | 1.6 |
| 69 | 71 | 25818 | 0.73958 |
| 70 | 80 | 29091 | 1.1268 |
| 71 | 71 | 25818 | 0.8875 |
| 72 | 63 | 22909 | 0.88732 |
| 73 | 65 | 23636 | 1.0317 |
| 74 | 47 | 17091 | 0.72308 |
| 75 | 74 | 26909 | 1.5745 |
| 76 | 63 | 22909 | 0.85135 |
| 77 | 78 | 28364 | 1.2381 |
| 78 | 68 | 24727 | 0.87179 |
| 79 | 55 | 20000 | 0.80882 |
| 80 | 48 | 17455 | 0.87273 |
| 81 | 51 | 18545 | 1.0625 |
| 82 | 45 | 16364 | 0.88235 |
| 83 | 48 | 17455 | 1.0667 |
| 84 | 50 | 18182 | 1.0417 |
| 85 | 71 | 25818 | 1.42 |
| 86 | 48 | 17455 | 0.67606 |
| 87 | 35 | 12727 | 0.72917 |
| 88 | 51 | 18545 | 1.4571 |
| 89 | 69 | 25091 | 1.3529 |
|  | **Total deaths from COVID-19 in 89 days according to the Ministry of Health of Iran: 7057** | **Sum Total Estimation of the number of cases based on the mortality rate obtained by the ANN-BTA network (0.275):2566200** | **Arithmetic mean daily basic reproduction number:1.0411** |

**Table 14. The effect of non-factor of government's order interventions on the total squared error of ANN-BTA network of the government with the mortality rate output of [0.1, 0.4].**

| Network error type | Mortality rate[0.1, 0.4] |
|---|---|
| MSE tr With all input factors | 0.0080 |
| MSE ts With all input factors | 0.0082 |
| MSE tr non-factor of government's order interventions | 0.0081 |
| MSE ts non-factor of government's order interventions | 0.0074 |

**Table 15. The superiority of the present study's method compared to the other methods.**

| Parameters | Proposed approach | Previous research |
|---|---|---|
| Precise and reliable data | Does not needs precise and reliable data | Needs precise and reliable data |
| Processing possibility with low and noisy data | Processing low and noisy data is possible | Processing low and noisy data is not possible |
| The discovery process of epidemiological knowledge | Discovery process of epidemiological knowledge does not need an expert epidemiologist | Discovery process of epidemiological knowledge needs an expert epidemiologist |

herd immunity time. That is, whenever it is necessary to extract vital knowledge from seemingly insignificant data, deterministic methods such as formulas 2 and 3 cannot be held accountable at all. Because these methods only produce reliable results when dealing with deterministic data of which their accuracy is proven and suffice for calculation. But the discovery of knowledge from noisy data and the limited number, is of capabilities of machine learning methods, an example of which, along with its efficiency and optimization, will be introduced in the present study to create an epidemiological model.

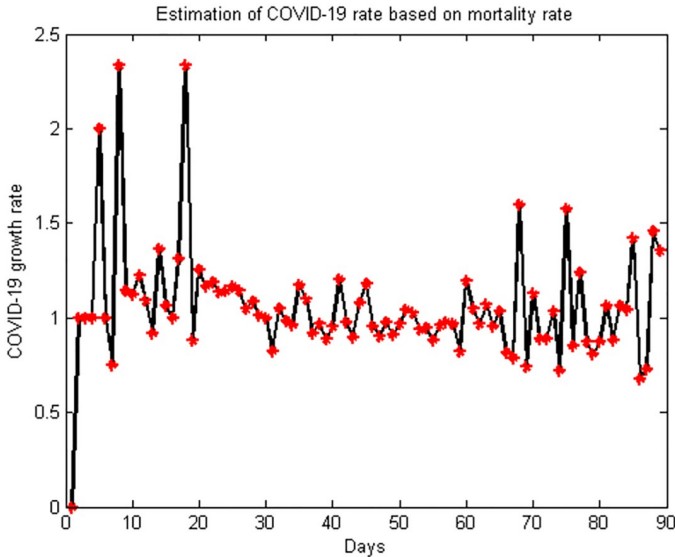

**Fig 13. COVID-19 growth rate in the period of February 19, 2020 to May 18, 2020 in Iran.**

## Supporting information

**S1 Data.**
(XLSX)

**S2 Data.**
(XLSX)

**S3 Data.**
(XLSX)

## Acknowledgments

We would like to thank the Meteorological Organization (IRIMO) for providing meteorological data. Also, we do appreciate the endless efforts of **Mr. Javad Azodi** assisting the proofreading of the present study.

## Author Contributions

**Conceptualization:** Mouhamad Bodaghie, Farnaz Mahan, Leyla Sahebi, Hossein Dalili.

**Data curation:** Leyla Sahebi.

**Methodology:** Mouhamad Bodaghie, Farnaz Mahan, Leyla Sahebi, Hossein Dalili.

**Software:** Mouhamad Bodaghie.

**Supervision:** Mouhamad Bodaghie.

**Writing – original draft:** Mouhamad Bodaghie.

**Writing – review & editing:** Mouhamad Bodaghie.

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
