## [Decision Letter · Decision Letter 0]

6 Sep 2021

PONE-D-21-24801A Novel Epidemiological Method based on Machine Learning Algorithms for Estimating the Number of Patients, Growth Rate, Mortality Rate and Herd Immunity Time Related to COVID-19 Disease in IranPLOS ONE

Dear Dr. Bodaghi,

Thank you for submitting your manuscript to PLOS ONE. After careful consideration, we feel that it has merit but does not fully meet PLOS ONE’s publication criteria as it currently stands. Therefore, we invite you to submit a revised version of the manuscript that addresses the points raised during the review process.

Please consider all comments of all reviewers including reviewer 1

We look forward to receiving your revised manuscript.

Kind regards,

Ahmed Mancy Mosa, Ph.D.

Academic Editor

PLOS ONE

Journal Requirements:

3. Please amend your Methods section to provide URLs for where the specific data in this study can be accessed.

4. Please update your submission to use the PLOS LaTeX template. The template and more information on our requirements for LaTeX submissions can be found at http://journals.plos.org/plosone/s/latex.

"No, The funders had no role in study design, data collection and analysis, decision to publish, or preparation of the manuscript."

6.In your Data Availability statement, you have not specified where the minimal data set underlying the results described in your manuscript can be found. PLOS defines a study's minimal data set as the underlying data used to reach the conclusions drawn in the manuscript and any additional data required to replicate the reported study findings in their entirety. All PLOS journals require that the minimal data set be made fully available. For more information about our data policy, please see http://journals.plos.org/plosone/s/data-availability.

7. PLOS requires an ORCID iD for the corresponding author in Editorial Manager on papers submitted after December 6th, 2016. Please ensure that you have an ORCID iD and that it is validated in Editorial Manager. To do this, go to ‘Update my Information’ (in the upper left-hand corner of the main menu), and click on the Fetch/Validate link next to the ORCID field. This will take you to the ORCID site and allow you to create a new iD or authenticate a pre-existing iD in Editorial Manager. Please see the following video for instructions on linking an ORCID iD to your Editorial Manager account: https://www.youtube.com/watch?v=_xcclfuvtxQ.

Reviewers' comments:

Reviewer's Responses to Questions

**Comments to the Author**

1. Is the manuscript technically sound, and do the data support the conclusions?

Reviewer #1: Yes

Reviewer #2: Yes

Reviewer #3: Yes

2. Has the statistical analysis been performed appropriately and rigorously? 

Reviewer #1: Yes

Reviewer #2: Yes

Reviewer #3: I Don't Know

3. Have the authors made all data underlying the findings in their manuscript fully available?

Reviewer #1: Yes

Reviewer #2: Yes

Reviewer #3: No

4. Is the manuscript presented in an intelligible fashion and written in standard English?

Reviewer #1: Yes

Reviewer #2: Yes

Reviewer #3: No

5. Review Comments to the Author

Reviewer #1: In this research, authors wish to estimate the morbidity, growth rate, and mortality rate of COVID-19 over a three-month period. The research is interesting, however, the proposed methodology existed in the literature and thus the novelty of the research is not sufficient to be published in PLOS ONE. Furthermore, the discussions and also the analysis of the manuscript need to be further improve in order to be published in PLOS ONE. So I suggest to give the author a chance to submit after doing additional research to improve the contributions of the research work. Authors may also consider the following comments for the revision work:

1. The format of the references need to be improve. For the literature review, only a paragraph is no sufficient. Author should refer to more ISI / Scopus journal, instead of online sources.

2. Authors need to identify the research gap and highlight the contribution(s) of the research work in the manuscript to shows the novelty of the research.

3. Authors should compare the proposed techniques with the state-of-arts to further prove the contribution(s)/novelty of the research work. There are also no justifications on why the proposed method being used instead of the state-of-arts?

4. More scientific reasoning should be added in the experimental results' explanations. There are lots of results tables shown in the text, however, very less discussion on those results being made. The purpose of having the results need to be clear.

5. The format of the tables need to be improved. The current one is a bit difficult to read. I will suggest author to highlight the significant one, e.g. those highest / lowest etc. and provide explanations on that.

Reviewer #2: Title of the scientific research: The researcher should use an attractive, eye-catching, short and easy-to-memorize title that motivates the reader to peruse the research.

The research summary is one of the most important parts of the scientific research, through which the reader is acquainted with the research and knows what it contains without having to read it in full.

Reviewer #3: The result is not convincing. Because it is clear for everyone age has a significant role in morbidity and mortality of covid.

Due to time limitation of publishing the study, the result is not practical.

The authors stated that “the role of quarantine policies implemented by the Iranian government was insigniﬁcant concerning the mortality rate” how this result has been concluded.

This type of studies need consultancy of an epidemiologist who is active in disease epidermis.

STRUCTURE OF REFERENCES

The result is not convincing. Because it is clear for everyone age has a significant role in morbidity and mortality of covid.

-Due to time limitation of publishing the study, the result is not practical.

-The authors stated that “the role of quarantine policies implemented by the Iranian government was insigniﬁcant concerning the mortality rate” how this result has been concluded.

-This type of studies need consultancy of an epidemiologist who is active in disease epidermis.

- REFERENCES writing structure in the text need to be corrected.

I don't have any idea about the technique of modeling.

6. PLOS authors have the option to publish the peer review history of their article (what does this mean?). If published, this will include your full peer review and any attached files.

Reviewer #1: No

Reviewer #2: No

Reviewer #3: No

---

## [Author Response · Author response to Decision Letter 0]

18 Oct 2021

A)Response to editor

q1:Please ensure that your manuscript meets PLOS ONE's style requirements, including those for file naming. 

a1:Required changes were made, including naming files based on PLOS ONE style.

q2: We suggest you thoroughly copyedit your manuscript for language usage, spelling, and grammar. If you do not know anyone who can help you do this, you may wish to consider employing a professional scientific editing service. 

a2:An overview of English grammar was done by a specialist in the field. The modifications are done in the new version compared to the previous version.

q3: Please amend your Methods section to provide URLs for where the specific data in this study can be accessed.

a3:In method section and dataset, the access URL was added and the followings were added to the same section:

The study data could be accessed through the below mentioned address:

https://www.dropbox.com/sh/t224son1u1z5hzq/AAB1GADIi6UdhwUI_C7GLggha?dl=0.

q4: Please update your submission to use the PLOS LaTeX template. The template and more information on our requirements for LaTeX submissions can be found at http://journals.plos.org/plosone/s/latex.

a4:The new version of the study was prepared according to PLOS ONE LaTeX template.

q5: Thank you for stating the following financial disclosure: 

"No, The funders had no role in study design, data collection and analysis, decision to publish, or preparation of the manuscript."...

a5:There were no funders funding the study. This is mentioned in the acknowledgement section of the study.

q6:In your Data Availability statement, you have not specified where the minimal data set underlying the results described in your manuscript can be found. ...

a6: In method section and dataset, the access URL was added and the followings were added to the same section:

The study data could be accessed through the below mentioned address:

https://www.dropbox.com/sh/t224son1u1z5hzq/AAB1GADIi6UdhwUI_C7GLggha?dl=0

B)Answer to First Reviewer

q1:The format of the references need to be improve. For the literature review, only a paragraph is no sufficient. Author should refer to more ISI / Scopus journal, instead of online sources.

a1:Most of the references were corrected and invalid ones were deleted; ISI/Scopus references were substituted in Review of the Related Literature (Related Works). They were revised according to journals format.

q2:Authors need to identify the research gap and highlight the contribution(s) of the research work in the manuscript to shows the novelty of the research.

a2: In the introduction, the following explanations were added to identify the research gap and to identify research contributions and research innovations. The numbering of the references and formula are observed in the main text of the study(pp:2-6).

If at a specific point in time (t), we define the percentage of "patients" in the population with the symbol Pt, the rate of disease transmission with the symbol Ct, and the infectious strength of the disease as a function of f (I) are shown. The formula (..) shows these three factors (ref43):

f(I,t)=C_t⋅P_t

The prevalence of the disease and its spread in society, based on the above relationship, is always moving towards a balanced pattern. From the point of view of epidemiology, two types of balanced patterns can be defined for the diseases.

 • Disease-free balance in which society is - certainly or almost - disease-free and stable in this respect. Example: At present conditions, for smallpox a disease-free balance can be defined in human society. 

• Endemic balance where the disease is endemic (or local) in the community and the occurrence of new cases follows a fixed and expected pattern (ref43).

In an epidemic, this balance is completely upset. Before the advent of Covid-19, the pattern of disease-free balance could be defined for society. But with the onset of the first cases, the pattern lost its balance and an epidemic occurred. This scenario will eventually lead to an endemic equilibrium pattern. But it is very important when and at what cost this balance will be achieved (from the perspective of morbidity and mortality) (ref44), the type and timing of control measures and interventions will be very important and effective at the end of this story. Estimation of baseline product number (R0) is important and valuable in monitoring the status of an epidemic until an epidemiological equilibrium is achieved (ref45). One of the most important and practical indicators used to show the pattern of expansion of Covid-19 is the baseline product number or (R0) nought R in which heterogeneities have been seen in different societies with the same conditions and parameters (ref46). By definition it is attributed to sensitive people who have been afflicted by contact by the disease people. This index basically indicates how contagious an infectious a given disease is (ref45). Concerning epidemiological concepts, the primary case is the person who transmits the disease to a small number of people in the community. This case has transferred the disease to all people during its infectious period. These people are called secondary cases. Each secondary case transmits the disease to other susceptible individuals. This cycle repeats itself periodically and the disease spreads in the community. Direct calculation of the basic reproduction number is possible when in a society with a very sensitive population, the number of people infected with the disease is carefully tracked from the initial case. In this case, the basic reproduction number can be calculated directly by calculating the average number of people that each specific patient can infect (ref48). The severity of the disease transfer is not constant at all, and is continuously changing. Therefore, its estimation at the beginning, middle and end of the epidemic can be completely different, but the difference in the same conditions in different societies may be due to a weakness in the model for extracting this index. Given that the basic reproduction number by definition should be calculated based on the course of the disease in a very sensitive community, so the symbol R0 can be used only in calculations or estimations at the beginning of the epidemic. Over time pass the reduced number of susceptible people in society is denoted by the symbol Rt or, for simplicity, by the symbol R. Theoretically, basic reproduction number depends on three factors:

 The risk of transmitting the disease per contact depends on two factors: the type of disease and the type of contact. It was first stated that Covid-19 disease is transmitted through direct contact with infected people and surfaces. Observing physical distance in socializing with others can be reduced. Estimates of this risk in routine contacts are estimated at 10 to 20% (ref….). but later the danger of infected surfaces was estimated so low that put the estimation results under the question (ref…).

 The average number of contacts per person per time unit, which depends entirely on population density, the frequency and manner of travel in the community, people's culture and social indicators can be modified with measures such as restricting or prohibiting meetings, reducing travel numbers, reducing the amount of interactions, isolation of patients or people suspected of having the disease, and quarantine of seemingly healthy individuals.

 The average period of infectivity, which, although is considered a stable biological indicator, can be reduced by therapeutic interventions and the administration of effective antibiotics. Estimates for Covid-19 disease show that the course varies from 14 days in mild cases (on average 5 days before symptoms and 9 days after) to several tens of days in severe and long cases (ref….-Ref ….). 

In addition to direct calculation, which is not always easily possible, it is possible to estimate the basic reproduction number based on mathematical models or learning methods such as neural network by using scientific assumptions and analyzing the time trend of occurrence. 

Many epidemics, including the recent Covid-19 epidemic, are possible due to the high percentage of asymptomatic subclinical patients and the expected low numbers in the care system, especially in cases where disease care is generally passive. Identifying primary and secondary cases and calculating the basic reproduction number directly is not easily possible. However, using learning methods and models such as neural networks, it is possible to approximate the basic reproduction number with a reasonable amount of basic error (ref….-Ref…).

On the other hand, neural networks combined with approaches with meta-heuristic algorithms in their learning algorithm can work more efficiently in approximating the basic reproduction number, which the present study seeks to prove experimentally. Previous approaches to basic reproduction number used a simple method to approximate it, based on the average age of infection with an infectious agent, which is more applicable to diseases that the number is expected to be relatively high for (ref…): 

R_0=1+L/A

In the formula (..), L is the average life expectancy of the population and A is the average age at the time of infection. Obviously, this formula does not have the necessary efficiency and proper mechanism to calculate quarantine conditions that have different ratings. 

Another previous approach to estimating the basic reproduction number is to use the rate of disease growth in the population taking into account the latency and infectivity rates of the disease. The basic reproduction number can be estimated by this method using three factors: 1) average incubation period of the disease 2) average period of pathogenesis 3) disease growth in the community. The following formula can be used for calculation (ref…)

R_0=K^2 (LD)+K(L+D)+1

In the above mentioned formula, L is the mean latency period, D is the average pathogenicity period, and K is the disease growth rate at the logarithmic scale. To calculate K, we can use Formula 3, where Y represents the number of patients and t represents time:

K=[Ln⁡(Y_t/Y_0 ) ]/t

This formula also has the problem of relation () which does not provide a mechanism for calculating social interventions and its effect on basic reproduction number.

Today, artificial intelligence approaches, especially machine learning techniques for modeling epidemiological data, have become increasingly common in the literature. These methods have the potential to improve our understanding of health and the opportunities for therapeutic and preventive interventions far beyond past techniques.

The reason for the superiority of artificial intelligence algorithms is that not only they have a high computational speed in the face of large and heterogeneous data, but also they can approximate high-order nonlinear data relationships with high computational complexity, thereby providing new knowledge; such as in epidemiological forecasting of diseases. Hence, they can achieve what the previous methods could not, because they do not have the computational ability with optimal time for this volume of data and in some cases, do not have suitable computational tools for some data (such as descriptive and fuzzy data). For example, when it comes to calculating the impact of government interventions through quarantine control of COVID-19 disease, finding data relationships and discovering knowledge from data that is not clearly articulated (such as the type of quarantine, social groups subject to quarantine, etc.) the previous methods of obtaining the prevalence rate or R number require a clear definition of such data.

Therefore, in general concerning the inefficiency of practices such as (fomo2), (fomo3) for estimating the production rate which can be used to predict the number of infected cases, the followings are suggested:

 These formula are only useful for extracting the basic production number and cannot be generalized to other indices like morbidity rate, prediction of morbidity and/or prediction of the time of herd immunity.

 They work comprehensively on deterministic data. As such they lack possible and efficient flexibility concerning non-deterministic data (probabilistic or fuzzy) and the combination of deterministic and non-deterministic data.

 For the existing noise in data, they do not produce reliable responses.

 They only produce reliable responses when there is a linear relation between data (variables) and given responses. In addition, they are not applicable for non-linear relations.

 Addition of any data after a specific response is produced requires remodeling which in turn contributes to increase in computational complexity.

 Computational complexity of these practices questions their efficiency for macro datasets. For example, the computational complexity of formula (Fomo2) at best mode is of 0(n2) degree. 

The present study presents a combined neural network algorithm with BTA algorithm to create a supervised machine learning model to estimate the basic reproduction number with taking into account the quarantine conditions at different levels. The error of estimating the basic reproduction number of the proposed algorithm in the present study is compared with combining the neural network with the PSO algorithm as another learning algorithm. 

Compared with machine learning based method, Due to limitations in defining their parameters as well as limitations in data type coverage, the conventional methods are not efficient in calculating epidemiological indices such as productivity rate index, mortality rate, predicting the number of cases and deaths, and even herd immunity time. That is, whenever it is necessary to extract vital knowledge from seemingly insignificant data, deterministic methods such as formulas () and () cannot be held accountable at all. Because these methods only produce reliable results when dealing with deterministic data of which their accuracy is proven and enough for calculation. But the discovery of knowledge from noisy data and the limited number, is of capabilities of machine learning methods, an example of which, along with its efficiency and optimization, will be introduced in the present study to create an epidemiological model.

The contributions of the present study are as follow:

 Proposing a model based on artificial intelligence to estimate the basic production number, predict the number of cases, mortality rate, the number of deaths and the time of herd immunity during the pandemic of COVID-19 virus.

 Combining artificial neural network with human based intelligence algorithm of BTA (Bus Transportation Algorithm) for utilization in the mentioned artificial intelligence model.

 Comparison of artificial neural network hybrid model combined with BTA algorithm, with neural network hybrid model combined with PSO algorithm in estimating the basic production number, predicting number of cases, mortality rate, mortality prediction and herd immunity time during the pandemic of COVID-19, and providing the experimental proof of the model proposed in the present study.

In section 2, in addition to an overview of recent studies aimed at proposing epidemiological models based on artificial intelligence, the existing gaps in related literature were addressed. Section 3 elaborates on the materials and methods; in section 3.1. a new epidemiological model based on artificial intelligence is introduced; in section 3.2. after a brief introduction of Bus Transportation Algorithm, the practice of combining artificial neural network with BTA is explained; section 3.3. refers to research data and the way it was accessed; section 3.4 describes the initial experimental quantification in the introduced hybrid neural network. In Section 4, the results of the model will be explained in details after implementation in the MATLAB environment and compared with a similar method. This section includes subsections for estimating the basic production number, predicting the number of cases, mortality rates, the number of deaths, and the time of herd immunity during the pandemic of COVID-19. Section 5 is devoted to discussion of the results and finally Section 6 presents the overall conclusion of the present study.

q3: Authors should compare the proposed techniques with the state-of-arts to further prove the contribution(s)/novelty of the research work. There are also no justifications on why the proposed method being used instead of the state-of-arts?

a3:Regarding the inefficiency of the existing methods, the following items were added to the introduction section in the new version(pp:4-5):

Therefore, in general concerning the inefficiency of practices such as (fomo2), (fomo3) for estimating the production rate which can be used to predict the number of infected cases, the followings are suggested:

1. These formula are only useful for extracting the basic production number and cannot be generalized to other indices like morbidity rate, prediction of morbidity and/or prediction of the time of herd immunity.

2. They work comprehensively on deterministic data. As such they lack possible and efficient flexibility concerning non-deterministic data (probabilistic or fuzzy) and the combination of deterministic and non-deterministic data.

3. For the existing noise in data, they do not produce reliable responses.

4. They only produce reliable responses when there is a linear relation between data (variables) and given responses. In addition, they are not applicable for non-linear relations.

5. Addition of any data after a specific response is produced requires remodeling which in turn contributes to increase in computational complexity.

6. Computational complexity of these practices questions their efficiency for macro datasets. For example, the computational complexity of formula (Fomo2) at best mode is of 0(n2) degree. 

Regarding the efficiency of the present study's method, the following items were added to the method section in the new version(pp:13-14):

Regarding the efficiency of the proposed model in the present study the following points can be counted:

1. The method not only is useful for extracting the basic production number, but also it can be generalized to other indices such as mortality rate, prediction of the number of mortality, and/or prediction of herd immunity time.

2. The method not only works on deterministic data, but also has possible efficiency flexibility for non-deterministic data and, combination of deterministic and non-deterministic datasets.

3. Although there is noise in data, reliable responses are produced (the existing noise is detected).

4. If there is linear and nonlinear relation between data and a specific response, the responses are reliably produced.

5. Addition of any data after a given response is produced, does not require remodeling which contributes not to high computational complexity.

6. The computational complexity of the method has required efficiency for macro datasets.

Regarding the comparison with one of the existing methods, the following items were added to the results section in the new version (although in both the old and the new version, comparisons were made between the artificial intelligence method of the present study (BTA) and the artificial intelligence methods previously proposed. There is a strong reason for the efficiency of the present study's method)(pp:8-9): 

Instead of using a model of a particular formula with specific variables, which can only represent the linear relationship between epidemiological variables, a learning model can describe not only the linear and nonlinear relationships of the data, but also can work on deterministic and non-deterministic data (such as fuzzy data) for extracting epidemiological indicators such as: productive rate, predicting the number of infected cases, mortality rate, predicting the number of deaths and finally the time of herd immunity. This model is a learning cycle that can learn more with each rotation and thus improve its error rate over time. The learning cycle of the epidemiological model is illustrated in Figure (1). This cycle has four stages, each of which is described in detail below. 

Experimental data to be used in learning systems:

Benefiting from previous experiments is an indispensable part of a learning system; where we are fully aware of the input and output data (such as basic production number, prediction of the number of infected cases, mortality rate, prediction of the number of deaths and the time of herd immunity) and we can consider such experiments as the basis of our system. In the case of COVID_19 the previous epidemics' data like MERS and EBOLA can be considered as the basis. The results of the (ref 49) could be a proper basis for a learning system when no data is available for basic production number, prediction of the infected cases, mortality rate, prediction of the number of deaths and the time of herd immunity. Therefore, if there are any available datasets it is logical to make use of them.

Selection of learning system (such as neural network):

A learning system must have appropriate flexibility regarding the data. The learning process must take place with due pace to be able to show required reaction. The architecture of a learning system is usually shaped through test and trial. In the present study the selected learning system is an artificial neural network discussed in section (ss2).

Selection of optimal learning algorithm (such as BTA):

Any given learning system requires an optimized learning algorithm. The pace of a given learning system highly depends on this algorithm. On the other hand the value of learning system's error is also dependent on the algorithm's efficiency. In the present study, BTA was used for learning in the neural network which is explained in details in section (SS2).

Data processing in the learning system and output:

In each cycle, some outputs are produced that can be used as the input in the next cycle. The process continues to reach an acceptable value of error to make the system more reliable.

q4:More scientific reasoning should be added in the experimental results’ explanations. There are lots of results tables shown in the text, however, very less discussion on those results being made. The purpose of having the results need to be clear.

a4: In the new version, the following items have been added in the explanation of the tables in the results section to explain the scientific reasons for the experimental results(pp14-31): 

According to the proposed epidemiological model based on machine learning, which was discussed in detail in the previous section, a hybrid neural network was designed with BTA and PSO learning algorithms. The structure of the designed neural network can be seen in Figure () and like other neural network structures, its structure in terms of the number of hidden layers and etc. has been obtained through trial and error. Figure () shows the final structure of the ANN-BTA network. . Regarding the structure of ANN-PSO network, the same structure has been preserved and it's the only substitute learning algorithm for BTA and PSO; this was presented to compare the proposed method of the present study i.e. ANN-BTA with a similar network, which proves its efficiency in similar problems. After the implementation of ANN-BTA and ANN-PSO networks in MATLAB environment, estimation of basic reproduction number and mortality rate due to CIVID-19 virus based on available data for 89 days in the time period of February 19, 2020 to May 18, 2020 in Iran was investigated. Relying on the two indicators of basic reproduction number and mortality rate, the number of cases and the number of deaths can be estimated or even predicted. After predicting the number of patients, the time of herd immunity can also be predicted.

(Section 4.1. was added and some changes were made in the place of the tables and their readability.)

4.1. Estimation of the basic reproduction number and, Estimating the number of cases, mortality rate, number of deaths and time of herd immunity without effective intervention the number of COVID-19 cases in the period of February 19, 2020 to May 18, 2020 in Iran.

In an artificial neural network, the output is reliable when the value of MSE in training and experimental samples is minimum. Both ANN-PSO and ANN-BTA neural networks had the lowest network error in the range of [1,1.2] in estimating the basic reproduction number, that confirms that the basic reproduction number of COVID-19 patients should be searched in this range. . Tables (tbl) and (tbl) show the ANN-BTA and ANN-PSO error rates in the ranges of [0.8,1], [1,1.2], [1.2,1.4] and [1.4,1.6], respectively. The point to consider is when the basic reproduction number is estimated higher than 1.4, in which the network error is greatly increased. Comparing the two tables, we find out that the proposed designed network's errors (ANN-BTA) are less in all ranges. This goes back to neural network design. Because part of the network error is related to the learning algorithm that the neural network benefits from. Considering the reduction of error in all the evaluated ranges, it can be concluded that ANN-BTA has learned better and as a result has obtained better results in network tests.

Table (tbl100): Total mean square error (MSE) of training and experimental data in the best implementation of ANN-BTA network for estimating basic reproduction number related to COVID-19 patients in the ranges of [0.8,1], [1,1.2], [1.2,1.4 ] And [1.4,1.6]

Table (tbl100): Total mean square error (MSE) of training and experimental data in the best implementation of ANN-BTA network for estimating basic reproduction number related to COVID-19 patients in the ranges of [0.8,1], [1,1.2], [1.2,1.4 ] And [1.4,1.6]

For more accurate estimation of basic reproduction number and based on the proposed method, the interval [1,1.2] was limited and more limited, and after several steps of interval limitation based on the minimum value of MSE, the network reached its lowest error at 1.45 (tbl1000 ). This value would be called the basic reproduction number and the value obtained by ANN-BTA network. The minimum value of MSE for estimating basic reproduction number in ANN-PSO network was 1.65 (2000 tbl). In both ANN-BTA and ANN-PSO networks, these intervals can be narrowed down to achieve lower MSE values.

Table (tbl1000) The MSE value of ANN-BTA network after interval limiting [1,1.2] and obtaining the lowest error rate in estimating basic reproduction numberand

Table (tbl2000) The MSE value of ANN-PSO network after interval limiting [1,1.2] and obtaining the lowest error rate in estimating basic reproduction numberand

Regarding the estimation of mortality rate among COVID-19 patients, it is interesting that both ANN-PSO and ANN-BTA neural networks had the lowest network error in the range of [0.1,0.4] and this confirms that the mortality rate among those with COVID-19 should be sought in this area. Tables (tbl1) and (tbl2) show the ANN-BTA and ANN-PSO error rates in the ranges of [0.1,0.4], [0.4,0.7], [0.7,1] and [1,1.3], respectively. The point to consider is when the mortality rate is estimated to be 1, in which case the network error increases significantly.

Table (tbl1) Total mean square error (MSE) of training and experimental data in the best implementation of the ANN-BTA network for estimating the mortality rate of COVID-19 patients in the ranges of [0.1,0.4], [0.4,0.7], [0.7, 1 ], [1.1.3], [1.3,1.7] and [1.7,2]

ANN-BTA network for estimating the mortality rate of COVID-19 patients in the ranges of [0.1,0.4], [0.4,0.7], [0.7, 1 ], [1.1.3], [1.3,1.7] and [1.7,2]

In order to more accurately estimate the growth rate based on the proposed method, the interval [0.1,0.4] was limited and narrowed, and after a few steps of interval limitation based on the minimum value of MSE, the network reached its lowest error rate of 0.275 (10000 tbl). This value can be called the basic reproduction number value obtained by ANN-BTA network. The minimum value of MSE for estimating mortality rate in ANN-PSO network was also obtained at 0.275 (20000 tbl). In both ANN-BTA and ANN-PSO networks, these intervals can be narrowed to achieve lower MSE values.

Table(tbl10000) The MSE value of ANN-BTA network after interval limiting [1,1.2] and obtaining the lowest error rate in estimating mortality rate

Table(tbl20000) The MSE value of ANN-PSO network after interval limiting [1,1.2] and obtaining the lowest error rate in estimating mortality rate

Tables (). Estimated number of patients with COVID-19 based on the mortality rate of patients obtained by ANN-BTA network in Table () (ie 0.275) and the mortality declared by the Ministry of Health of Iranduring the time period of February 19, 2020 to May 18, 2020 in Iran. Also, based on this estimation, the basic reproduction number is calculated daily.

At the bottom of the table () the arithmetic mean of the basic reproduction number in 89 days is shown. It can be clearly seen that the basic reproduction number obtained by ANN-BTA (i.e. 1.045) mentioned in tbl1000 is very close to the arithmetic mean of basic reproduction number and in 89 days (i.e. 1.0411). 

Table (). Estimation of basic reproduction number and number of patients per day in the period of February 19, 2020 to May 18, 2020 in Iran based on the mortality rate obtained by the ANN-BTA network and the mortality rate declared by the Ministry of Health of Iran.

The following was added to conclusion part(pp:33-34):

Although, at first glance the findings of the study may be obvious and to some extent approved-due to the information that has been obtained since the beginning of the pandemic of COVID-19 virus- the mortality rate, basic production number and/or prediction of the time of herd immunity, and the effectiveness of quarantine policies in a country like Iran could be elaborated on. However, the importance of the findings of the present study come forth when we are aware that they were obtained through the available data such as meteorological or fuzzy ones with much noise and errors on how to apply quarantine in Iran; second, the range of these data was so limited. Thus, the present study shows that the epidemiological model based on the proposed machine learning algorithm (with little available noisy data) could be used, and perhaps persuading health policymakers to tighten quarantine policies for further executions in future.

On the other hand, creative works such as combining artificial neural network with human intelligence algorithm based on BTA (bus transportation algorithm) to be used in an epidemiological model based on the current study's machine learning method called ANN-BTA network, showed more efficiency and performed better compared to the other advanced methods such as ANN-PSO in proposed epidemiological model; numerous tests approved the above mentioned issue.

The results of the present study showed that, compared with machine learning based method, Due to limitations in defining their parameters as well as limitations in data type coverage, the conventional methods are not efficient in calculating epidemiological indices such as productivity rate index, mortality rate, predicting the number of cases and deaths, and even herd immunity time. That is, whenever it is necessary to extract vital knowledge from seemingly insignificant data, deterministic methods such as formulas () and () cannot be held accountable at all. Because these methods only produce reliable results when dealing with deterministic data of which their accuracy is proven and enough for calculation. But the discovery of knowledge from noisy data and the limited number, is of capabilities of machine learning methods, an example of which, along with its efficiency and optimization, will be introduced in the present study to create an epidemiological model.

q5:The format of the tables need to be improved. The current one is a bit difficult to read. I will suggest author to highlight the significant one, e.g. those highest / lowest etc. and provide explanations on that.

a5: The tables' format has been changed to improve readability and comprehension. The required amendements were answered in previous comment.

C)Response to Second Reviewer

q1:Title of the scientific research: The researcher should use an attractive, eye-catching, short and easy-to-memorize title that motivates the reader to peruse the research.

The research summary is one of the most important parts of the scientific research, through which the reader is acquainted with the research and knows what it contains without having to read it in full.

In the new version, the title of the paper was changed as follows to shorten and include the general techniques and objectives of the article:

Neo-Epidemiological Machine Learning based Method for COVID-19 Related Estimations

As a summary of the study, the following items were added to the introduction in the new version(pp:6): 

In section 2, in addition to an overview of recent studies aimed at proposing epidemiological models based on artificial intelligence, the existing gaps in related literature were addressed. Section 3 elaborates on the materials and methods; in section 3.1. a new epidemiological model based on artificial intelligence is introduced; in section 3.2. after a brief introduction of Bus Transportation Algorithm, the practice of combining artificial neural network with BTA is explained; section 3.3. refers to research data and the way it was accessed; section 3.4 describes the initial experimental quantification in the introduced hybrid neural network. In Section 4, the results of the model will be explained in details after implementation in the MATLAB environment and compared with a similar method. This section includes subsections for estimating the basic production number, predicting the number of cases, mortality rates, the number of deaths, and the time of herd immunity during the pandemic of COVID-19. Section 5 is devoted to discussion of the results and finally Section 6 presents the overall conclusion of the present study.

D)Response to Third Reviewer

q1and q2: The result is not convincing. Because it is clear for everyone age has a significant role in morbidity and mortality of covid.

Due to time limitation of publishing the study, the result is not practical.

a1 and a2:Of course, it is evident that some results are clearly known, but in the present study the focus was on proposing a neo-epidemiological machine learning based model resorting to dataset based on narrow bulk of accessible data and the initial circumstances at the beginning of the COVID-19 outset. Therefore, the method was more elaborated on and enriches; the results were also made more practical to cover the aim of the study.

q3:The authors stated that “the role of quarantine policies implemented by the Iranian government was insigniﬁcant concerning the mortality rate” how this result has been concluded.

a3:Generally, the concept of quarantine policies was not the same as what was implemented in Europe and America. So it was tangible and obvious. Therefore, it is meant that the Iranian policies were insignificant e.g. the public offices were open, travel restrictions were not observed and etc.

q4:This type of studies need consultancy of an epidemiologist who is active in disease epidermis.

a4: Required consultancy was done by Dr. Leila Sahebi. Hence, some epidemiological concepts were modified and revised. Like basic reproduction number which were refered in the previous version as infection rate.

q5: STRUCTURE OF REFERENCES.

a5: Most of the references were corrected and invalid ones were deleted; ISI/Scopus references were substituted in Review of the Related Literature (Related Works)

The following were added to Related Works(pp:6-8):

Researchers in [18] presented a three-step machine learning strategy for classifying risk across a given country based on countries reporting COVID-19 data. They classified four risk groups based on risk of transmission (cases of COVID-19 per million population), risk of death (deaths per million population from COVID-19) and risk of inability to use a COVID-19 test (COVID-19 test per million population). Corona per million population) for countries worldwide. The four risk groups were labeled: "low", "medium-low", "medium-high" and "high". They used the Stack of Gradient Boosting, Decision Tree, and Stack of Support Vector Machine methods in their classification. The method used by these researchers proved the efficiency of machine learning methods for classifying the mentioned epidemiological indices.

In [19], a machine learning method inspired by the least squares model (SIMPLS) has been developed to predict mortality at hospitals. Prediction accuracy is randomly assigned to training sets and validation. SIMPLS-based model is developed by isolating rescued people from survivors. The method was able to predict hospital mortality in patients with COVID-19 with moderate predictive strength (Q ^ 2 = 0.24) using training and validation kits, which in turn revealed high accuracy (AUC> 0.85).

As in [20] it was declared that the basic reproduction number has been the spotlight of reported information during the last six months. The value of the reproduction number has been used by different range of practitioners in medical science, hospitals, political decision makers and public media to justify the strategies used to take the COVID-19 pandemic under control. The study elaborated on the effectiveness of political interventions using the basic reproduction number of COVID-19 across Europe. The researchers proposed a SEIR epidemiological model with a time varying reproduction number, identified through the use of machine learning method. As they report the basic reproduction number was 4.2±1.69, with maximum value of 6.33 in Germany and Netherlands during the early outbreak of COVID-19. The value decreased to 0.67±0.18, with minimum value of 0.37 and 0.28 by May, 10, 2020 in Hungary and Slovakia. A strong correlation was found between passenger air travel, walking, and transit mobility and the effective basic reproduction number with a time delay of 17.24±2.00 days. The proposed new dynamic SEIR model avails flexibility to estimate outbreak control and the exit strategy to inform the political decision makers and identify global safety solutions.

In [21] asserted that long-term forecast of COVID-19 pandemic assists health authorities to determine the transmission features of the virus and take effective steps in prevention and control strategies in advance. They proposed Dynamic-Susceptible-Exposed-Infective-Quarantined (D-SEIQ) model; resulting from due modifications applied to Dynamic-Susceptible-Exposed-Infective-Recovered (SEIR) and integrating machine learning based parameters optimization under epidemiological rational constraints. The model was used in order to predict the long-term cumulative numbers of COVID-19 cases in China from January, 27, 2020. Reports from three different regions in China were selected for model evaluation. The results approved the effectiveness of the model in simulating and predicting the trend of COVID-19 outbreak. The researchers declared that integrated approach of pandemic and machine learning models could accurately forecast the long-term trend of the COVID-19 outbreak. In addition, the parameters of the proposed model were insightful for analyzing COVID-19 transmission and the effectiveness of related interventions in China. 

In [22], the researchers investigated the disease spreading behavior along with the basic reproduction number benefiting from susceptible-infected-recovered (SIR) model. They simulated the disease transmission activity by using Monte Carlo simulation and analyzed it benefiting from the design of experiment and neural network. The investigated systems were considered as discrete cells for allocating the agents i.e. system population. Varied sizes and population densities were used to observe the finite size effect, while the infectious period was varied to observe its influence on disease transmission dynamics. Results indicated that the number of agents in each phase as a function of time depended on the whole parameters. The main plot suggested that the basic reproduction number maintained with the increased system size; increased to some extent by increasing the density, and a significance increase was observed with the increase in infectious period. For establishing the relationship among parameters, the researchers resorted to neural network and found out the optimized network architecture to be at 3-28-9-1. They also made use of residual plot analysis to confirm the quality of obtained data. The validity of using multiple modeling/analysis techniques i.e. Monte Carlo, design of experiment and neural network, as the supplementary essential tools for modeling the dynamics of SIR disease spreading scheme, was approved with high level of accuracy for data prediction.

Although all of above mentioned studies used machine learning in some way to estimate epidemiological indices, the disadvantage of all of these studies is the lack of a general epidemiological model that utilizes machine learning-based abstraction to estimate epidemiological indices. In fact, all the mentioned models used machine learning in the heart of their classic model, which does not lead to filling all the basic gaps of the classic model. For example, all of these models are non-generalizable to different types of epidemiological data and work for a specific type of data. But the abstraction on which the present study is based introduces a model that is highly generalizable to any type of data and space (both fuzzy and even quantum). This model has other capabilities that will be discussed after the model is introduced.

---

## [Decision Letter · Decision Letter 1]

12 Nov 2021

PONE-D-21-24801R1Neo-Epidemiological Machine Learning based Method for COVID-19 Related EstimationsPLOS ONE

Dear Dr. Bodaghi,

Thank you for submitting your manuscript to PLOS ONE. After careful consideration, we feel that it has merit but does not fully meet PLOS ONE’s publication criteria as it currently stands. Therefore, we invite you to submit a revised version of the manuscript that addresses the points raised during the review process.

Please consider all comments

We look forward to receiving your revised manuscript.

Kind regards,

Ahmed Mancy Mosa, Ph.D.

Academic Editor

PLOS ONE

Reviewers' comments:

Reviewer's Responses to Questions

**Comments to the Author**

1. If the authors have adequately addressed your comments raised in a previous round of review and you feel that this manuscript is now acceptable for publication, you may indicate that here to bypass the “Comments to the Author” section, enter your conflict of interest statement in the “Confidential to Editor” section, and submit your "Accept" recommendation.

Reviewer #1: (No Response)

2. Is the manuscript technically sound, and do the data support the conclusions?

Reviewer #1: Yes

3. Has the statistical analysis been performed appropriately and rigorously? 

Reviewer #1: Yes

4. Have the authors made all data underlying the findings in their manuscript fully available?

Reviewer #1: Yes

5. Is the manuscript presented in an intelligible fashion and written in standard English?

Reviewer #1: Yes

6. Review Comments to the Author

Reviewer #1: In this research, authors proposed an existing framework on different domain. The research is interesting, this paper is suggested to be published in PLOS ONE after authors have addressed the comments listed below:

1. Authors need to highlight the research gap and highlight the contribution(s) of the research work in the manuscript to shows the novelty of the research.

2. I will suggest author to perform a sensitivity analysis to make the results to be more convincing.

3. More critical discussion is needed on the results and discussion. The current one is not sufficient. E.g. why the small improvement is significant?

4. More recent references can be added to the manuscript.

7. PLOS authors have the option to publish the peer review history of their article (what does this mean?). If published, this will include your full peer review and any attached files.

Reviewer #1: No

---

## [Author Response · Author response to Decision Letter 1]

7 Jan 2022

Question1: need to highlight the research gap

and highlight the contribution(s) of the

research work in the manuscript to shows

the novelty of the research.

response1: On page 5 before listing the contributions of the present study the following items were added:

According to the above said issues and studying similar content, the general gaps existing in the body of literature can be listed as below:

1- All models estimating the basic production number need reliable and precise information.

2- The previous models estimating the basic production number have worked on narrow data, and do not cover large data and those with noise.

3- Parameters of the previous models estimating the basic production number are not flexible and do not work with the same data.

4- Discovery process of epidemiological knowledge in the previous models estimating the basic production number is not possible without the mastery and knowledge of an expert epidemiologist.

 Question2:I will suggest author to perform a

sensitivity analysis to make the results to be

 response2: Due to the fact that extracting results with large data requires time processing based on artificial intelligence systems, so adding another evaluation index for the present study requires a lot of time, but in any case the following comparison table was added to the results section: Table 17 shows the superiority of the present study's method in terms of processing with accurate data, the possibility of processing with low data volume and noisy data, and the process of epidemiological knowledge discovery. The present method not only does not require accurate data and has the ability to go through the process of predicting epidemiological processes with accessible data, but also the process of discovering epidemiological knowledge in the present study does not require expert knowledge and with any level of knowledge the epidemiological knowledge can be understood.

Table 17: The superiority of the present study's method compared to the other methods

Question3: More critical discussion is needed on the

results and discussion. The current one is

 not sufficient. E.g. why the small

RESPONSE3: The following content was added to the discussion section:

The importance of the improvements that have taken place in the results lies in the fact that these results are obtained with data that the accuracy of all this data is not verifiable and in other words we do not encounter reliable data. Therefore, slight improvements in the results of the present study with data obtained in the early COVID-19 pandemic period promise that when reliable data are not available, we can rely on the learning method of the present study, which over time due to learning, the system's answers are more accurately and complete. On the other hand, the improved obtained answers are not necessarily obtained with the help of an expert epidemiologist, and with any level of epidemiological knowledge, knowledge can be extracted. Another problem that adds to the value of the responses obtained is that by processing large volumes of data that are definitely noisy, a response worthy of attention with little improvement over previous methods is obtained.

Question4:More recent references can be added to the

 RESPONSE4: The following references have been added to the article:

---

## [Editor Report · Decision Letter 2]

2 Feb 2022

Neo-Epidemiological Machine Learning based Method for COVID-19 Related Estimations

PONE-D-21-24801R2

Dear Dr. Bodaghi,

We’re pleased to inform you that your manuscript has been judged scientifically suitable for publication and will be formally accepted for publication once it meets all outstanding technical requirements.

Kind regards,

Ahmed Mancy Mosa, Ph.D.

Academic Editor

PLOS ONE
---

## [Editor Report · Acceptance letter]

20 Apr 2022

PONE-D-21-24801R2 

Neo-Epidemiological Machine Learning based Method for
COVID-19 Related Estimations 

Dear Dr. boudaghi:

I'm pleased to inform you that your manuscript has been deemed suitable for publication in PLOS ONE. Congratulations! Your manuscript is now with our production department. 

Kind regards, 

on behalf of

Dr. Ahmed Mancy Mosa 

Academic Editor

PLOS ONE